# Strategic Behavior in Two-sided Matching Markets with Recommendation-enhanced Preference-formation

**Stefania Ionescu**
Department of Informatics
University of Zürich
Andreasstr. 15 , Zürich, 8050
ionescu@ifi.uzh.ch

**Yuhao Du**
Department of CS and Engineering
University at Buffalo
Buffalo NY, 14260
yuhaodu@buffalo.edu

**Kenneth Joseph**
Department of CS and Engineering
University at Buffalo
Buffalo NY, 14260
kjoseph@buffalo.edu

**Anikó Hannák**
Department of Informatics
University of Zürich
Andreasstr. 15 , Zürich, 8050
hannak@ifi.uzh.ch

## Abstract

Two-sided matching markets have long existed to pair agents in the absence of regulated exchanges. A common example is school choice, where a matching mechanism uses student and school preferences to assign students to schools. In such settings, forming preferences is both difficult and critical. Prior work has suggested various prediction mechanisms that help agents make decisions about their preferences. Although often deployed together, these matching and prediction mechanisms are almost always analyzed separately. The present work shows that at the intersection of the two lies a previously unexplored type of strategic behavior: agents returning to the market (e.g., schools) can attack future predictions by interacting short-term non-optimally with their matches. Here, we first introduce this type of strategic behavior, which we call an adversarial interaction attack. Next, we construct a formal economic model that captures the feedback loop between prediction mechanisms designed to assist agents and the matching mechanism used to pair them. Finally, in a simplified setting, we prove that returning agents can benefit from using adversarial interaction attacks and gain progressively more as the trust in and accuracy of predictions increases. We also show that this attack increases inequality in the student population.

## 1 Introduction

In *two-sided matching markets*, agents are partitioned in two disjoint sets (e.g., students and schools) and want to get paired with agents from the other set for future bilateral exchanges (e.g., a student learns at a school and the school provides instruction to the student) [26]. In such markets, agents typically report their preferences over potential matches and a *matching mechanism* uses those preferences to produce an *assignment*, i.e. a pairing between agents. Because forming one's preference is hard, agents benefit from external recommendations.

Since predictive models have become increasingly accessible, reliable, and trusted [17], they also became more frequently used to inform preference-formation in matching markets. One such example is school choice. After the introduction of the No Child Left Behind Act, in 2001, students from low-performing schools were allowed to transfer to better-performing schools. However, initially, only a few students took advantage of this opportunity, partly because it was hard for parents to assess which

37th Conference on Neural Information Processing Systems (NeurIPS 2023).

school would improve their child's performance. In 2008, a content-based recommender system called SmartChoice was deployed for focus group participants; its goal was to help parents with the assessment by identifying the best schools based on predictions on the student's development at that school [27]. Another example is in refugee assignment, where refugees are matched with locations. Here, the preferences of locations over refugees are given by a machine learning (ML) model that predicts the level of integration success (e.g., measured by the probability of employment within 90 days) of an individual at a given location [5]. 'The Swiss government has recently implemented a randomized test to examine the performance of data-driven algorithms for outcome-based assignment' [3]. Similar approaches have been developed for other application domains such as labor market [24] and course allocation [15].

These examples show that using predictive models to inform the preferences of agents in matching markets is not a problem of the future, but rather one of the present. However, prior work has evaluated potential vulnerabilities *separately* in the matching mechanisms [10, 7, 2] and prediction-based recommendations [23, 21]. When considering the matching algorithm, one usually analyzes the incentives of individuals by, for example, asking whether the mechanism is *strategy-proof* (i.e. whether agents have an incentive to misreport their preferences). Similarly, there is a broad literature on attacks on recommendations [8, 23, 21] and predictive models which might face *poisoning* or *evasion* attacks (i.e. attacks which either inject fake data to trigger an unfaithful model or perturb the testing input to trigger a misclassification) [11].

In this paper, we argue that, in addition to considering vulnerabilities of the matching and prediction mechanisms independently, it is critical to also look at vulnerabilities in systems that *combine them together*. Specifically, we consider systems similar to those described above, i.e, where: (a) agents of one side (the *returning side*) come back to the matching market in subsequent rounds (e.g., schools), (b) agents have post-matching objectives (e.g., schools want to be prestigious and minimize their cost), (c) agents on the returning side *have the power to shape the quality of the interaction* with those they are assigned to via post-matching decisions (e.g, schools can increase the performance of students through extra-curricular preparations or integration programs), and (d) these interactions impact outcomes for the non-returning side, which in turn influence future predictions (e.g., predictions based on the outcomes of past students change the preferences of current students).

By making the feedback loop between the matching and prediction mechanisms explicit, we uncover a new type of strategic behavior: the returning side can attack the system by changing their interactions with their matches. Even though in the short-run it might be beneficial for both matched parties to have the best possible interaction (e.g., both locations and refugees want refugees to be employed as soon as possible), the returning side (e.g., locations) might want to sacrifice their utility in the current round for different future predictions (e.g., locations might postpone hiring difficult-to-integrate refugees so they will not be allocated similar refugees in the future). We call such post-matching strategic behavior—where agents sacrifice their short-term utility to trigger different long-term predictions—*adversarial interaction attacks*. Similarly, we call a system where agents cannot benefit from such attacks *interaction-proof*.

In this paper, we first build a formal model for repeated two-sided matching markets with prediction-enhanced preference formation. Second, we use this model to define *adversarial interaction attacks* and the optimization problem faced by returning agents. Finally, we create a simplified setting (in the main text, and an agent-based model in the appendix) to analyze when and by how much agents can benefit from interaction-strategic behavior. We show that, for some systems, (a) the returning agents have an incentive to use adversarial interaction attacks, (b) the utility gains obtained through such attacks increase as predictions become more accurate and trusted, (c) once a returning agent attacks, others have an incentive to implement more severe attacks, and (d) the non-returning agents are unevenly negatively affected by such attacks.

## 2   Other Related Work

*Adversarial Attacks in Recommendations.* From past literature in recommender systems (RS), the most similar adversarial attacks to the one we consider are *shelling attacks* (i.e., attacks where fake users and ratings are created to trigger different future recommendations). These works mainly focused on crafting adversarial examples [23, 21]. Differently, Christakopoulou and Banerjee [8] analyzed shelling attacks from a machine learning and optimization perspective by building on the

literature on poisoning attacks in classification tasks [16], and formulating the problem as a two-player general-sum game between an RS and an Adversarial Attacker.

However, our setting has some key differences. First, the returning agents usually cannot inject fake users (e.g., because there are public records of which students attended which schools). Instead, agents attack by adapting their actions. This distinction has both social implications (the non-returning agents experience a different outcome while the attack takes place) and economic implications (the returning agent implementing the attack sacrifices their short-term utility). Second, the existence of capacity constraints entails a rivalry for available seats (if a student is accepted at a school then another student is not accepted, and vice versa). Consequently, when attacking for being matched to a specific (type of) agent, there are two intents to do so: (a) rank higher in the preferences of agents who are of interest, and *(b) rank lower in the preferences of agents who are not of interest*. The latter is different from classical recommender system applications and is increasingly important when the returning agents cannot fully and freely express their preferences. Third, also because of competition, agents do not decide to whom they are allocated (not all students can get a place to their most preferred school); instead, they report their preferences to a matching market, which produces an assignment. Therefore, attacks in this setting are not only efficient if they change the most preferred option of a non-returning agent, but also if they change the ordering of options lower in the preference ranking.

*Strategic Behavior in Matching Markets.* As noted above, matching markets (MM) are widely used to pair agents based on their reported preferences; in such settings agents can behave strategically by, e.g., misreporting their preferences. Prior work shows that users find and implement such profitable manipulations which leads to congestion and inefficiencies [7]. Thus, strategy-proofness is a common desiderata in real-world application domains; e.g., in the Boston school choice system, this triggered a transition from the Boston to the Deferred Acceptance mechanism [1]. In this paper we analyse strategic-interacting, which differs from strategic-reporting as it is not an attack on the mechanism alone, but rather on its combination with the prediction-based preference-formation process. When proposing predictive modelling for refugee assignment, Bansak et al. [5] distinguished between the modeling phase (when the predictive model is built) and the matching phase (when the assignment is produced). We generalise and extend this framework by also considering the phase when the paired agents interact and new data is produced. Moreover, we build on prior literature in economics when creating the model (for formalising the matching phase [25]) and when analyzing the system (for unilateral deviations and Nash equilibria [22]).

*Using Simulations to Understand Long-Term Effects.* The machine learning community has seen an increasing use of simulation to study the interaction between users and recommendations. RecSim [12] provides an environment that naturally supports sequential interaction with users. Mansoury et al. [19] proposed a method for simulating interactions between users and RSs to study the impact of the resulting feedback loop on the popularity bias amplification. Bountouridis et al. [6] built a framework called SIREN to study how RSs will impact users' news consumption preference in the long term. Similarly, scholarship within MM used simulations to understand the effect of different design choices. Erdil and Ergin [10] developed an experiment in order to compare the performance of two alternative matching algorithms in the school choice setting. In the context of online dating, Ionescu et al. [13] proposed an agent-based model (ABM) to test the effects of different platform interventions in reducing racial homogamy. At the intersection of ML and two-sided markets, Malgonde et al. [18] built an ABM to test the effects of introducing a two-sided RS within a complex adaptive business system. Similar to this previous work, we create a simplified setting to model a complex socio-technical system. We use both theoretical analysis (in the main text) and simulations (in the appendix – to confirm that the results carry to more realistic settings). We, however, have a different goal: to understand how the characteristics of the market affect the incentives to use and effects of using adversarial interaction attacks.

## 3 Problem Formulation

Our model for the system is composed of three phases: (a) modeling, (b) matching, and (c) interacting. The modeling stage builds a predictive model that forecasts the interaction outcome of two agents if matched. During the matching phase, the non-returning agents use this model to inform their

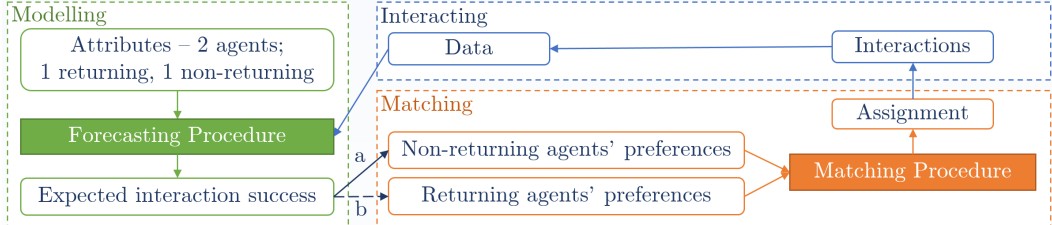

Figure 1: Overview of the interaction between different components of a system with a forecasting-based preference formation and a matching procedure. The interaction predictions can be used to inform the preferences of a. the non-returning side of the market (e.g. students), or b. the returning side of the market (e.g. locations).

preferences [1] ; next, a *matching algorithm* pairs each agent in the non-returning side of the market (student) with an agent in the returning side of the market. Finally, in the interacting stage, the agents – already paired according to the assignment obtained during matching – interact with each other (students attend the classes at their assigned school). This interaction will produce an outcome (e.g., SAT scores of students) that is kept as a record and used to inform future predictions. The entire system then repeats in a series of rounds with new non-returning agents each time, but with the same returning agents. Figure 1 shows an overview of all the phases and the interactions between them. In the remainder of this section, we formalise each of the stages and present the decision problem faced by the returning agents.

### 3.1   The Matching Stage

We start with the notation for the agents. Let $X^t$ be the set of non-returning agents to be matched at time (round) $t$, and $Y$ be the set of returning agents. Moreover, we denote by $\succ_x$ the preference of the non-returning agent $x \in X^t$ over the returning agents, $Y$. In other words, $\succ_x$ is a ranking over $Y$. Similarly, $\succ_y$ is the preference of $y \in Y$ over $X$. These preferences could be either unweighted (i.e., the agent only knows the ordering of their options) or weighted (i.e., the agent also knows how much more they prefer each potential match over another). In the weighted case, the preference $\succ_x$ has an associated weight function $w_x : Y \to \mathbb{R}$ mapping each option $y$ to the strength $x$ wants to be matched to $y$.

A matching is a pairing of non-returning agents to returning ones, i.e., $\mu : X \to Y \cup \{\perp\}$. Here, $\mu(x) = \perp$ signifies that $x$ remained unassigned[2]. More generally, a matching procedure (or allocation rule) maps the preferences of agents to a matching. In other words, a matching procedure $M$ is a function mapping every $(\{\succ_x\}_{x \in X}, \{\succ_y\}_{y \in Y})$ to a matching $\mu$. We denote by $\mu^t$ the matching in round $t$, i.e. $\mu^t = M((\{\succ_x\}_{x \in X^t}, \{\succ_y\}_{y \in Y}))$.

### 3.2   The Interacting Stage

After being matched, the agents interact. For every agent pair $(x, y)$, there is a set of possible outcomes, depending on their interaction. Since $x$ is not returning to the market, we assume they will always prefer the best outcome for them. Therefore, the set of possible outcomes depends on the actions of the returning agent, $y$. We denote the set of outcomes $y$ could choose from when interacting with $x$ by $\mathcal{O}_y(x)$.

Depending on the resulting outcome, each agent has some value, cost, and utility. If $y$ chooses outcome $o$ when interacting with $x$ we denote by $v_y^o(x)$ their value, by $c_y^o(x)$ their cost, and by

---

[1]Alternatively, the predictive model could be used to inform the preferences of the returning agents. For simplicity, we chose not to extend our model to capture this alternative in the main text and only include it in the appendix. Moreover, here we use school choice as a running example; see the appendix for other examples.

[2]In simplified scenarios where each returning agent can be matched to at most one non-returning agent (e.g., due to capacity constraints), we can express the matching function as a function from $Y$ to $X \cup \{\perp\}$. More precisely, $\mu(y) = x$ when $x$ is matched with $y$ and $\perp$ if $y$ remains unmatched. Since this refers to the same pairing, we avoid introducing additional notation and will also use $\mu$ for the mapping of returning to non-returning agents in such simplified cases.

$u_y^o(x) = v_y^o(x) - c_y^o(x)$ their utility. We use the analogous notation for $x$. We also make the simplifying assumptions that (a) the value is symmetric, i.e. $v_y^o(x) = v_x^o(y)$, and (b) the cost of the non-returning side is null, i.e. that $c_x^o(y) = 0$ and $u_x^o(y) = v_y^o(x)$.

As an example, a school $y$ might have the choice set $\mathcal{O}_y(x) = \{80, 100\}$ since it could either invest in extracurricular preparation for $x$ (case in which $x$ scores around $100\%$ on state-administered tests) or not to invest (and $x$ scores around $80\%$). If school $y$ incurs a fixed cost of 5 when preparing a student, then for the outcome 80 both the student and the school have a utility of 80 while for the outcome 100 the student has a utility of 100 while the school subtracts the cost from the value, and thus has a utility of $100 - 5 = 95$.

### 3.3 The Modeling Stage

The prediction model at time $t$ is informed by the history of interactions until that time. We denote the history up to time $t$ by $H^t = \{(x, y, o, t') | y = \mu^{t'}(x), y \text{ interacted with } x \text{ at time } t' \leq t \text{ with outcome } o \in \mathcal{O}_y(x)\}$. The history thus records the agents matched so far, together with the time and outcome of their interaction.

The prediction algorithm uses the history as an input. Its output is a *hypothesis*, i.e., a function which maps a pair of agents to their expected interaction outcome. We use the usual statistical framework in learning theory and denote the hypothesis by a parameterized function $h_{\theta_t} : X \times Y \to \mathbb{R}$. The prediction is thus based on the parameter $\theta_t$ at time $t$, which is obtained by solving the optimization problem $\theta_t = \arg\min_\theta L(\theta, H^{t-1})$. Here, $L$ is the loss function.

The hypothesis predicts values for a potential interaction between a non-returning agent, $x$, and all the returning agents. Therefore, it induces a preference weight function and a ranking over the returning agents. Formally, $h_{\theta_t}(x, \cdot) : Y \to \mathbb{R}$ gives a weighted preference over $Y$, which we denote by $\succ_x^h$. By exposing agent $x$ to this ranking, their final preference might change. To capture this change, each agent $x$ has a *prediction integration function*, $h_{\theta_t}^x$, that transforms a (weighted) preference of $x$ into a new preference, $\succ_x'$, using the hypothesis-based ranking, $\succ_x^h$.

To continue the previous example, let us assume that student $x$ initially believes schools $A$ and $B$ are equally good for them, i.e. $\succ_x$ is $A \sim B$ with, say, a weight function $w_x(A) = w_x(B) = 80$. Assume the hypothesis predicts $h_{\theta_t}(x, A) = 100$ and $h_{\theta_t}(x, B) = 80$, which corresponds to the ranking $A \succ_x^h B$. The exposure of student $x$ to this prediction-based ranking changes their preference to a new one which lies between their original opinion, $\succ_x$, and the suggested one, $\succ_x^h$. For instance, the new preference could correspond to the weight profile obtained by averaging the original and the suggested ones. Then, the new preference, $h_{\theta_t}^x(\succ_x) = \succ_x'$, has the weight function $w_x'(A) = 90$ and $w_x'(B) = 80$ thus ranking $A \succ_x' B$.

### 3.4 The Decision Problem of Returning Agents

The decision problem faced by the returning agents when they choose how to interact with their matches can be formalised as a dynamic programming task over the infinite horizon. They choose the action that maximizes the sum of their utility now plus the expected discounted utility in the future if they take that action. Using the notation introduced in the previous section, the maximum expected-discounted utility of a returning agent, $y$, from time $t$ onward is [3]

$$U_y\left(\mu^t\right) = \max_{o \in \mathcal{O}_y(\mu^t(y))} \left(u_y^o\left(\mu^t(y)\right) + \beta \mathbb{E}\left[U_y\left(\mu^{t+1}\right) | (\mu^t(y), y, o, t) \in H^t\right]\right), \tag{1}$$

where we make the standard assumption in economics that agents steeply discount future utility and denote the discount factor by $\beta$. The *optimal strategy* of a returning agent is to take, at every round, the interaction leading to the outcome that maximizes the expected-discounted utility.

The straightforward interaction strategy is to always choose the outcome maximising the 1-step utility, i.e. choose $\arg\max_{o \in \mathcal{O}_y(\mu^t(y))}\left(u_y^o\left(\mu^t(y)\right)\right)$. When a returning agent uses this strategy, we say it *interacts truthfully*. Naturally, in general, the optimal strategy need not be the truthful one. The resulting gap leaves space for strategic interactions. We refer to strategies involving non-truthful

---

[3]To achieve this succinct problem formulation, we made some simplifying assumptions and notations (details in Subsection 2.3 from the appendix).

interactions as *adversarial interaction attacks* and call the systems in which agents cannot benefit from such attacks *interaction-proof*.

To give two simple examples, a system in which the preference formation is solely based on the attributes of schools (e.g., how far away the school is, what subjects are taught, what are the final examinations) and past data on interactions is not used would be interaction-proof. Similarly, a system that only uses the history of interactions before announcing the decision to introduce the ML-algorithm would also be interaction-proof. This is because, in both of these cases, the future expected utility is constant with respect to the choice of interaction. Thus, the utility from now onward is maximized by choosing the outcome giving the highest utility in the current round.

# 4  Simplified Setting

While useful for formalizing the definition of adversarial interaction attacks within a variety of contexts, the problem formulation proposed in the previous section is too general to allow for a tractable and insightful numerical analysis. Thus, we operationalize the framework by creating a simplified model. The main goal of such a model is to support both proving that returning agents can benefit from adversarial interaction attacks and understanding why and when this could be the case. The remainder of this section introduces the simplified model. To improve readability, we refer to returning agents as schools and non-returning agents as students. However, the model could be read through the lens of alternative applications (e.g., locations and refugees).

*Interaction.* We assume that there are two schools, $Y = \{A, B\}$, and two types of students: cheap, $C$, and expensive, $E$. Moreover, each school has the capacity to accept only one student. Any interaction between a school and a student leads to an outcome between zero and one (i.e., $\mathcal{O}_y(x) = [0, 1] \; \forall y \in Y, x \in \{C, E\}$). This outcome matches its value: $v_y^o(x) = o$. However, schools have different costs for a given outcome depending on the type of student. More precisely, cheap students require negligible specialized assistance, thus having a zero cost for any outcome, i.e. $c_y^o(C) = 0$, while expensive students require a cost of three quarters of the value of the chosen outcome, i.e. $c_y^o(E) = 3/4 \cdot o$. Thus, the resulting utility for a school $y \in Y$ is given by $u_y^o(C) = o$ and $u_y^o(E) = o/4$. Note that this always leads to non-negative utilities. To account for cases when returning agents incur large integration costs from accepting any match (e.g., for refugee assignment or the allocation of under-performing students [27, 5]) we also analyze the scenario of negative utilities. We do so by offsetting the cost by a constant $q > 1$, thus giving $u_y^o(C) = o - q$ and $u_y^o(E) = o/4 - q$. Unless otherwise specified, we consider the non-negative model for utilities (i.e., without the offsets).

*Preferences.* For simplicity, we assume students have a uniform random chance of preferring each school, and predictions are based on the outcome of the most recent interaction of the same-type student at the school. More precisely, a student $x$ will have a $1/2$ chance of having the preference profile $A \succ_x B$ and a $1/2$ chance of having the preference profile $B \succ_x A$. For the same student $x$, the prediction algorithm gives to each school $y$ a weight $w_x^h(y)$ equal to the last outcome of a same-type student at school $y$ based on the history of interactions[4]. For example, a student of type $E$ will start with a $50\%$ chance of thinking that $A \succ_E B$; if, according to the history, the most recent student of type $E$ who attended school $A$ had a worse outcome than the most recent student of type $E$ who attended school $B$ then an accurate prediction-based hypothesis will recommend $B \succ_E^h A$.

*Accuracy and trust.* The impact of predictions depends on their accuracy and trust, two metrics on which system designers usually try to improve on [20, 9]. To account for this, we model the accuracy via a parameter $\alpha$, which captures the probability that the recommendation is the one given by the most recent outcome with a same-type student. In the example above, the hypothesis will generate $B \succ_E^h A$ with probability $\alpha$ and the reversed ranking (i.e., $A \succ_E^h B$) with probability $1 - \alpha$. Similarly, the trust of students in the recommendations is given by $\theta$, which captures the probability that students adopt the recommended ranking as opposed to maintaining their original preference over the alternatives (i.e., $\succ_x'$ is $\succ_x^h$ with probability $\theta$ and $\succ_x$ with probability $1 - \theta$). The accuracy and trust are just extensions to help understand the changes in incentives to attack as the power of our

---

[4]In the absence of historical interactions between a type of student and a school we assume the outcome was 1. We believe this is the most sensible assumption as prior to introducing a prediction mechanism to inform preferences there is a limited impact of past outcomes on future assignments. As a result, schools would likely optimize the 1-step utility and interact truthfully.

prediction models improves. Since the final goal of predictions is to be accurate and trustworthy, for most of the analysis we will assume both $\alpha$ and $\theta$ are 1.

*Matching.* Matching mechanisms are complex procedures that, based on the preferences of all agents, assign students to schools. Serial dictatorship, Boston, and deferred acceptance are all examples of such mechanisms which were deployed in the school choice setting (see the appendix for the definition of each). While, in theory, schools have the chance of expressing their preferences, in practice, these preferences are often given by a lottery with some priority ordering (e.g. students with siblings at the same school have a higher priority at that school) [1]. Thus, the true preferences of schools usually have a limited impact on the final allocation. For the theoretical analysis, we consider random serial dictatorship (RSD), which (1) picks a random ordering of students and (2) assigns, in turn (according to this ordering), each student to the most preferred school that still has available places. In the appendix, we use simulations to investigate the effects of using alternative mechanisms.

*Analysis.* To see whether schools have an incentive to interact strategically, we compare the utility of one school, $y$, under two scenarios: (a) when both schools interact truthfully (i.e., pick outcome 1 for their interaction with all students) and (b) when $y$ interacts strategically at some level $l$ (i.e., picks outcome $1 - l$ where $l \in (0, 1]$ for its interaction with expensive students) while the other school interacts truthfully. If the expected utility of $y$ increases when it unilaterally interacts strategically, then $y$ has an incentive to deviate from interacting truthfully and adopt such a strategy. Using a game-theoretic framework, we say that schools are agents which can choose actions corresponding to *adversarial attacks at levels* $l \in \{0, 1/L, 2/L, \dots, 1\}$[5]: an attack at level $l$ means that the school always chooses the outcome $1 - l$ when interacting with expensive students and the outcome 1 with cheap students. Thus, $l = 0$ represents a truthful interaction. With respect to this set of actions, each school has a *best response* (i.e., an action that gives it the highest utility) given the actions of others. If both schools play the best response to the other's actions and, thus, none of them can change their level of attack to achieve a higher expected utility, we say the action profile is a *Nash Equilibrium.*

## 5 Results

The results section starts with three subsections, each titled based on its main takeaway. The first two focus on results under the assumption that prediction models have perfect levels of trust and accuracy, while the third subsection will investigate incentives to attack as those levels change. Since proofs are mostly mathematical formalizations of intuitive behavior, we reserve them for the appendix and solely provide the intuition within the main text. To position the results within the broader context, each of these three subsections ends with a paragraph discussing the implications of its results. We also include a final subsection explaining why we opted for a simple model and analysis and how it could be extended.

### 5.1 Not all systems are interaction-proof

We first consider a scenario where, at each timestep, there is one student of each type (i.e, $X^t = \{C, E\}$) and schools put sufficient weight on the future expected utility (i.e., $\beta$ is sufficiently large). In such a case, Theorem 1 states that schools can benefit from implementing an adversarial interaction attack. The proof (see appendix) compares the expected utilities under truthful and strategic interactions and shows that an attack at any level $l \neq 0$ produces an increase in expected discounted utility when $\beta > 2/3$.

**Theorem 1** *Assume $X^t = \{C, E\}$, schools are sufficiently forward-looking ($\beta > 2/3$), and predictions have perfect trust and accuracy (i.e., $\alpha = \theta = 1$). Then, schools have an incentive to deviate from truthfully interacting and implement adversarial interaction attacks.*

It is important to note that this takeaway is heavily based on the composition of the market (i.e., the number and type of students available). For example, Theorem 2 shows that when there are only expensive students, a strategy profile where all schools interact truthfully is a Nash Equilibrium. Similarly, Theorem 3 shows that, even when students induce a negative utility for the schools, as long

---

[5]Here, $L$ is an arbitrary large natural number. This makes the action space discrete, thus capturing the often-used assumption that agents do not have infinitesimal control over the outcomes. Instead, there is an atomic unit of change ($1/L$ in our case) they can make.

as they are of the same type and sufficiently many to guarantee that each school will be allocated a student, then schools still do not have an incentive to implement adversarial interaction attacks. The full proofs are included in the appendix. They formalize the intuition that, in such scenarios, schools either (a) prefer being allocated a student to remaining unmatched or (b) have the same type of allocated student disregarding their interaction outcomes.

**Theorem 2** *Assume predictions have perfect trust and accuracy (i.e., $\alpha = \theta = 1$). When there are only expensive students, a strategy profile where both schools interact truthfully is a Nash Equilibrium.*

**Theorem 3** *Assume predictions have perfect trust and accuracy (i.e., $\alpha = \theta = 1$). When all students induce the same negative utility for schools, and there are at least two students, a strategy profile where both schools interact truthfully is a Nash Equilibrium.*

Yet, there are also scenarios when forward-looking schools have an incentive to interact strategically with all available students, disregarding their type. One such instance is when there is one student giving schools a negative utility for any choice of action. This example is covered by Theorem 4. The main intuition behind the proof is that the attacking school has the opportunity to avoid being allocated any student, which gives it a better utility than being allocated one. If the weight given to future gains (i.e., $\beta$) is sufficiently large, then schools prefer to decrease their utility in the current round in order to maximize the chance of having a null utility in subsequent rounds.

**Theorem 4** *Assume predictions have perfect trust and accuracy (i.e., $\alpha = \theta = 1$). If there is only one student giving schools a negative utility for any outcome and schools are sufficiently forward-looking, then schools have an incentive to deviate from truthfully interacting and implement adversarial interaction attacks.*

These toy examples prove that it is not sufficient to analyze the matching mechanism in isolation from the preference-formation process. In particular, Roth [25] proved that RSD is a strategy-proof mechanism for both sides. However, our theorems show that a system containing the same mechanism is not necessarily interaction-proof, thus challenging the desirability of implementing such a mechanism.

### 5.2 Once a school implements an attack, the other school has an incentive to respond with more severe attacks

The prior analysis showed that, in some setups, schools have an incentive to interact strategically. It remains, however, unclear how a school would best respond to an adversarial attack of the other school. Theorem 5 states the best response of a school is to attack at the smallest level that is still higher than the current level of attack of the other school. The proof (see appendix) formalizes the intuition that schools want an attack that sacrifices their short-term utility the least, while still triggering the prediction mechanism not to recommend them to expensive students.

**Theorem 5** *Assume predictions have perfect trust and accuracy (i.e., $\alpha = \theta = 1$). When $X^t = \{C, E\}$ and schools are sufficiently forward-looking ($\beta > 4/5$), the best response of school A for a strategy B of attacking at a level $l$ is to attack at a level $\min\{1, l + 1/L\}$.*

This has a few important implications. First, it suggests that schools would likely not attack at high levels directly, thus having a limited impact on the welfare of expensive students. However, little by little, schools would attack at increasingly higher levels until both schools choose an outcome of $0$ (i.e., level of attack 1) for expensive students. Corollary 1, which is an immediate consequence of Theorem 5, states that the resulting strategy profile when both schools attack at a maximum level is a Nash Equilibrium. This equilibrium is, however, undesirable for both schools, as it is Pareto dominated by the initial choice of actions; i.e., if both schools interact truthfully, then both of their expected utilities are higher than when the schools attack at the maximum level.

**Corollary 1** *Assume predictions have perfect trust and accuracy (i.e., $\alpha = \theta = 1$). When $X^t = \{C, E\}$ and schools are sufficiently forward-looking, the strategy profile where both schools use adversarial interaction attacks at the highest level (i.e., $l = 1$) is a Nash Equilibrium. However, this is Pareto dominated by the strategy profile where all schools interact truthfully.*

Altogether, these results show the system resulting from introducing prediction-based recommendations to inform preferences within a matching market could encourage returning agents to engage in higher and higher levels of adversarial interaction attacks. This behavior ultimately leads to an equilibrium that is detrimental to all agents. For our example, it lowered the welfare (i.e., the average value of outcomes) of expensive students and reduced the expected discounted utilities of both schools. At a societal level, such a system could thus result in increased inequalities among the student population. Therefore, there is a need for a system with prediction and matching mechanisms that together lead to different interaction incentives for returning agents. As illustrated by the next two subsections, two potential solutions are to lower the impact of outcome-based predictions or use matching mechanisms that let schools express their true preferences.

### 5.3 Increases in trust in and accuracy of recommendations lead to higher discounted expected utilities under adversarial interaction attacks

Finally, we acknowledge the general trend of increasing the accuracy of and trust in prediction [20, 9]. Before incorporating prediction mechanisms, prior outcomes have a limited impact on preferences (e.g., from anecdotal experiences of isolated past students, or advice from counselors [27]). When predictions are made algorithmically, there is a higher potential for personalizing them; however, due to the high level of individuality, algorithms struggle to produce accurate predictions. This affects both the trust of students in recommendations and leads to a challenge for designers to improve accuracy. Theorem 6 shows that improving these parameters affects the impact of adversarial interaction attacks. The proof is based on the following observation: the expected gain in utility from using a strategic interaction is strictly increasing with both trust and accuracy.

**Theorem 6** *When $X^t = \{C, E\}$, the expected utility of a school using an adversarial interaction attack is a continuous function that increases monotonously in both accuracy ($\alpha$) and trust ($\theta$).*

Naturally, when predictions are disregarded by students (i.e., $\theta = 0$) or are random (i.e., $\alpha = 0.5$) it is best for schools to interact truthfully – see Theorem 7. When increasing the accuracy and trust, the utility under attacks increases (by Theorem 6) until adversarial attacks constitute a best response (as shown by Theorem 1). This means that, depending on the specifics of the system, such as the parameter $\beta$, for any given level of trust, there exists some maximum value of $\alpha$ where truthful interaction is still the optimal strategy. The proof of this statement can be formalized using the intermediate value theorem and the three aforementioned results.

**Theorem 7** *When $X^t = \{C, E\}$, the truthful strategy profile is a Nash Equilibrium under predictions with 50% accuracy ($\alpha = 0.5$) or null levels of trust ($\theta = 0$) .*

In short, the above analysis shows that, even if adversarial interaction attacks do not currently increase the expected utility, they might if we introduce prediction mechanisms or improve their trust and accuracy. This underlines the importance of re-evaluating incentives when considering such changes. Moreover, it shows that limiting the impact of outcome-based predictions could, in turn, limit the gains of returning agents from interacting strategically.

### 5.4 Discussion

Our paper (1) created an economic model to allow for a general definition of adversarial interaction attacks and (2) simplified the setting to prove these attacks could indeed be attractive for returning agents. We underline that our goal was to illustrate the emerging feedback loop and show the importance of considering it when prediction algorithms inform preferences. Thus, we favored a simple model together with a basic theoretical analysis. As such, the current analysis can be extended in various ways, such as by using more realistic prediction and matching mechanisms (MM), accounting for more diverse agents, defining and analyzing incentives for undetectable interaction attacks, and extending the space of strategies.

Perhaps the main question for future work is whether adversarial interaction attacks remain attractive in more realistic settings. As a first step to address this question, we extended the simple setting presented above to a more realistic agent-based model. In particular, this model uses k-Nearest Neighbors [14, 4] to predict future outcomes, compares multiple MM (including Deferred Acceptance and Boston), and increases the number of agents. The simulation results both confirm the takeaways

from the theoretical analysis and provide some additional insight. First, they show that different MM are more susceptible to adversarial interaction attacks depending on the distribution of students with respect to their attributes. Second, they suggest that allowing schools to express their true preferences over students (instead of using lotteries) can, in some cases, eliminate the incentives of using adversarial attacks. The exception is when utilities are negative and there are fewer students than places at schools. Third, it illustrates the inequality between students: as suggested by the theoretical results, the welfare of students decreases when schools interact strategically, and this decrease is mostly borne by the expensive students. We provide details on the agent-based model and visualizations for the simulation results within the appendix; to ease reproducibility, the code is available on GitHub[6]

## 6 Conclusion

Predictive models are increasingly used to inform preference-formation in matching markets (MM). However, the robustness of the prediction mechanism under adversarial attacks and of MMs under strategic behavior are usually investigated separately. In the present work, we extend existing models by including the interacting stage in which agents, matched by the MM, interact with each other, thus generating new training data. Doing so makes the feedback loop between the prediction model and the MM explicit and uncovers a new type of strategic behavior: the agents that return to the market in subsequent rounds can deviate from the most profitable interactions in the current round in order to attack the predictions and matchings of future rounds.

In a simplified setting, we prove that returning agents could indeed have an incentive to interact strategically. Doing so is *more effective* as predictions get *better* in terms of accuracy (and trust). In alignment with previous work on recommendations [20], this suggests we should look beyond accuracy when designing prediction mechanisms for such systems. Moreover, we find that, when returning agents choose to adopt this attack, it has perverse consequences for the utility of all agents. In particular, it both lowers the overall utility *and* increases inequality among non-returning agents (students). These issues reflect how strategic behaviors of social institutions (schools) cause social inequalities independent of individual potential. Altogether, this work indicates that both aspects of the matching and prediction mechanisms are key in developing robust systems and sets the framework for a dialog between the ML and MM communities.

## Acknowledgments and Disclosure of Funding

The authors would like to thank Vlad Gavrila, Atri Rudra, Nicolò Pagan, and the anonymous reviewers for their valuable feedback. KJ and YD gratefully acknowledge the support by the NSF award IIS1939579, with partial support from Amazon.

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
