# Strategic Behavior in Two-sided Matching Markets with Recommendation-enhanced Preference-formation – Supplementary Material –

**Stefania Ionescu**
Department of Informatics
University of Zürich
Andreasstr. 15 , Zürich, 8050
ionescu@ifi.uzh.ch

**Yuhao Du**
Department of CS and Engineering
University at Buffalo
Buffalo NY, 14260
yuhaodu@buffalo.edu

**Kenneth Joseph**
Department of CS and Engineering
University at Buffalo
Buffalo NY, 14260
kjoseph@buffalo.edu

**Anikó Hannák**
Department of Informatics
University of Zürich
Andreasstr. 15 , Zürich, 8050
hannak@ifi.uzh.ch

## 1   Organization of Supplementary Material

This document provides supplementary material for several points in the main text. We, thus, organize it roughly in the order referenced in the body of the paper. First, Section 2 presents additional information regarding the problem formulation. This includes a discussion on the link between the model and the real-world applications, a description of the most used matching mechanisms, some notes on assumptions and notation for the dynamic programming equation, and some examples of interaction-proof systems. Second, the following section presents the proofs of the theorems in the main text. Third, Sections 4 and 5 present the agent-based model used for the virtual experiment and the simulation results, respectively. This document ends with a discussion of extensions and directions for future work.

## 2   Problem Formulation

### 2.1   Extensions and Application Domains

#### 2.1.1   Human-based Predictions

Note that our model is agnostic to where the predictions are formed (e.g., in the school choice setting, it is agnostic to whether a guidance counselor [14] or a statistical model is making the predictions [16]). Previous work shows that not only are algorithms more accurate than humans at making predictions [6], but also, in some settings, they are more trusted by participants [12]. Therefore, a transition from human-based to algorithmic-based predictions also leads to increased accuracy of and trust in predictions. Paired with our findings from the main text, this means that introducing an, e.g., ML algorithm to make predictions in such settings could increase the gains obtained from using adversarial interaction attacks. As we will see later in the supplementary material, this is also confirmed by the simulation results.

37th Conference on Neural Information Processing Systems (NeurIPS 2023).

### 2.1.2 Refugee Assignment

As mentioned in the introduction, refugee assignment is another application domain using a matching market with prediction-enhanced preference formation [4, 2]. In this case, refugees are the non-returning agents, and locations are the returning agents. There is, however, a distinction between how predictions are used in school choice and in refugee assignment. In school choice, predictions help students form their preferences, while in refugee assignment, they are used to generate the preferences of locations. In Figure 1 from the main text, arrow (a) is used to show predictions that are used to inform the preferences of the non-returning agents (similarly to the school choice setting), and arrow (b) is used to show predictions that are used to inform/dictate the preferences of the returning agents (similarly to the refugee assignment setting).

Our problem formulation can be straightforwardly extended to account for this second type of influence. To do so, we note that the hypothesis, $h$, also induces a ranking over the non-returning side of the market; $h_{\theta_t}(\cdot, y) : X \to \mathbb{R}$ gives a weighted preference over $Y$, which we denote by $\succ_y^h$. By considering a *prediction integration function* for the returning side too, $h_{\theta_t}^y$, we obtain a way of transforming a (weighted) preference of $y$ into a new preference, $\succ_y'$, using the hypothesis-based ranking for the returning agent, $\succ_y^h$. In the case of refugee assignment the integration function simply returns the hypothesis-based ranking, meaning that the locations completely follow the predictions.

### 2.1.3 Other Application Domains

In this subsection, we explain how our model extends to three other application domains. First, Paparrizos et al. [15] proposed a recommender system (RS) that accurately suggests job transitions based on prior data of employees who changed jobs. In this case, the employees who want to change jobs are the non-returning side, and the employers offering jobs are the returning side. The interaction consists of an application/transition assessment (e.g., probation period); such an interaction is successful if the applicant did successfully transfer to the new job position and unsuccessful if the applicant did not. If employers face (a) a cost of assessing candidates and/or (b) legislation imposing restrictions on which applicants to hire, then they could have an incentive to interact strategically.

Second, also in the context of job recommendations, Liu et al. [11] suggested a prediction-based RS to help college students find jobs. In this case, recommendations are based on the similarity between current and past students and the feedback obtained from past students who attended certain jobs. Similarly to before, students are the non-returning agents, and employers are the returning agents. The interaction outcome is decided based on the attributes of the student.

Third, Kurniadi et al. [10] proposed an RS that suggests courses to students based on performance predictions. For this application domain, students are the non-returning agents, and courses are the returning agents. The outcome of the interaction is the result of the student (e.g., GPA or exam grade). Therefore, this application is similar to the school choice setting – the only difference being that courses replace schools.

## 2.2 Matching Mechanisms

*SD and RSD.* One straightforward example of a matching algorithm is the serial dictatorship mechanism. The students are considered in some order (e.g., in the order of their GPA, or at random), and each student gets allocated to the first school in their ranking that still has available places. Random serial dictatorship (RSD) refers to the version of the mechanism when students are ordered at random. For our simulation (later in the supplementary material), we use SD to refer to the version of the mechanism when students are ordered by their entry level of knowledge. [1]

*Boston.* The Boston mechanism proceeds by rounds:

- *Round 1.* Students apply to their first choice according to their preference. Schools accept the most preferred applicants (i.e., the students ranked highest), subject to capacity constraints.

---

[1]For SD and RSD, the preferences of schools are not used directly. Instead, the mechanism designer makes an implicit assumption about the preferences of the schools, depending on the student ordering they are using. For example, if students are considered in the order of their GPA, then the implicit assumption is that schools prefer students with higher GPA, while if the ordering is at random, the assumption is that schools are indifferent between which students they are matched with.

- *Round k.* Currently unassigned students apply to their k-th choice (if such a choice exists in their ranking). Schools accept the most preferred applicants from that round, such that they do not exceed their remaining places.
- *End.* The procedure terminates when either (a) there are no more unassigned students or (b) the unassigned students do not have any school left to apply to.

*DA.* The Deferred Acceptance (DA) mechanism is similar to Boston. However, depending on the round, there are some key differences:

- *Round 1.* Schools **tentatively** accept the most preferred applicants.
- *Round k.* Currently unassigned students apply to their most preferred school **to which they did not apply before**. Schools consider both the applicants from the current round and the tentatively accepted students from before and choose the students they prefer the most. This forms a new set of tentatively accepted students.

## 2.3 Decision Problem - Assumptions and Notation

To achieve the succinct problem formulation for the maximum expected discounted utility of a returning agent (i.e., equation (1) in the main text), we made some simplifying assumptions and notations. First, we assumed $\mu^t$ maps each returning agent to exactly one non-returning agent. Note that this formula can be extended to the general case where each returning agent is assigned a (possibly empty) set of non-returning agents. The only non-trivial step in doing so is to determine the utility of a non-returning agent over a set of interactions (e.g., the utility of a school when interacting with two students could be the sum of the utility in the interaction with each). Second, we did not expand on how the assignment in the next iteration is formed. As a reminder, this is a complex process that depends on a variety of factors, such as the interactions of the other schools, the attributes and original preferences of the students arriving in the next round, the effect of the new history on the RS, the way students integrate the recommendations, and the preferences of the other schools. All these variables produce a (believed) distribution over the possible allocations in the next round. Consequently, the expected value is taken over this distribution.

## 2.4 Examples of Interaction-proof Systems

To give two simple examples, a system in which the preference formation is solely based on the attributes of schools (e.g., how far away the school is, what subjects are taught, what are the final examinations) and past data on interactions is not used would be interaction-proof. Similarly, a system that only uses the history of interactions before the decision to introduce the ML-algorithm would also be interaction-proof. This is because, in both of these cases, the future expected utility is constant with respect to the choice of interaction. Thus, the utility from now onward is maximized by choosing the outcome giving the highest utility in the current round.

# 3 Proofs of Theoretical Results

This section of the supplementary material is dedicated to the proofs of the theoretical results. We start by underlying and discussing the assumptions, together with introducing some additional notation. Next, we introduce four lemmas, which will aid in proving the theorems. These lemmas are kept general such that they are used throughout different settings (e.g., for general values of accuracy of and trust in predictions, various types of students). The remaining subsections match the order and titles of the ones within the Results Section of the main text. They provide the proofs of the respective theorems.

## 3.1 Assumptions and Additional Notation

*Strategic Behavior.* We underline that, as mentioned in the paper, the set of considered strategies is restricted. More precisely, we only consider strategies where schools always misinteract in the same way with undesired students (i.e., for any timestep and actions of the other agents). There are two reasons for this. First, we believe this set of actions is more realistic. Since strategic interacting with protected non-returning agents in such settings is ultimately discriminatory and could have serious

social consequences, we believe it to be unreasonable that returning agents would openly discuss their strategies. In addition, since the set of outcomes is, in general, private information, it is very hard to infer and especially prove that an agent interacted strategically. Altogether, this means that tit-for-tat strategies (i.e., strategies where a returning agent threatens to strategically interact in response to another agent using such a strategy) are unrealistic in our scenario. Second, we chose this restricted set of strategies to reduce the complexity of the analysis. Although our setting bears similarities with repeated prisoner dilemma, the actions of returning agents are based on predictions, matching, and initial preferences of non-returning agents. As such, a formal game-theoretic formulation of this setting would require stochastic games, thus significantly increasing the complexity of the analysis. Since our goal was to clearly prove that agents could benefit from adversarial interaction attacks and open the door for discussions between researchers across domains, we opted to avoid the complexity of analyzing a stochastic game and favored a more simple and realistic alternative.

*Range for Parameters.* For parameters $\alpha$, $\beta$, and $\theta$ we consider typical ranges. First, we say that the discount range for future utility is non-negative and less than 1, i.e., $\beta \in [0, 1)$. Second, we assume the accuracy of predictions is a number between 0.5 and 1, i.e., $\alpha \in [0.5, 1]$. This assumption is standard when the output space is binary, since a model with accuracy lower than 0.5 can be replaced by a corresponding one which consistently swaps predictions. In our case, a model that simply outputs the reversed ranking would perform better. Finally, the trust of non-returning agents in predictions is, naturally, a number between 0 and 1, i.e. $\theta \in [0, 1]$

*Additional Notation.* To keep upcoming proofs brief and clear, we introduce some additional notation. First, since we solely consider stationary strategies (i.e., strategies where schools will always choose a fixed outcome for undesirable students, for any timestep and past actions of the other school), we denote strategy profiles by $\sigma = (\sigma_A, \sigma_B) \in [0, 1] \times [0, 1]$, where $\sigma_y$ is the outcome school $y$ will choose whenever interacting with an undesired (e.g., expensive) student. Second, since the utilities do not depend on schools (i.e., $u_A^o(x) = u_B^o(x)$) we omit the subscripts (i.e., denote the utility simply by $u^o(x)$). Moreover, since schools always choose the outcome 1 for desirable students, we omit the superscript for their generated utility (i.e., for a desirable student $x$ the utility generated for the school is $u(x)$). Third, since the prediction only changes with variations of the most recent interaction outcome of undesired students, $H_y$, at each school, $y$, we denote the history by $H = (H_A, H_B) \in [0, 1] \times [0, 1]$. Fourth, we denote the expected discounted utility of a school $y$ under the strategy profile $\sigma$ and history $H$ by $U_y(\sigma|H)$. If, moreover, we know that the current matching is $\mu$, then the expected discounted utility is $U_y(\sigma|H, \mu)$.

## 3.2 Auxiliary Results

The first lemma computes the expected discounted utility of one school when both have and will use the same strategies. It applies both when there are two or only one students available. For the former, the utilities to be matched with each type of student are $u_1$ and $u_2$. For the latter, we say that the utility of a school to be matched to the student when following the commonly used strategy of interaction is $u_1$, and the utility of remaining unmatched is $u_2$. Under these conditions, the lemma states that the expected discounted utility is $\frac{u_1 + u_2}{2} \cdot \frac{1}{1-\beta}$.

**Lemma 1** *Let $u_1$ and $u_2$ be either (a) the utilities from each respective student (when there are two students per round) or (b) the utilities from the existing student and remaining unmatched (when there is only one student per round). Then, when both schools have and will treat each available student the same way (i.e., $\sigma_A = \sigma_B$ and $H_A = H_B$), their expected discounted utility is $\frac{u_1 + u_2}{2} \cdot \frac{1}{1-\beta}$.*

**Proof:** Let $U$ be the expected discounted utility of a school, say $A$, when both schools use the same strategy and had the same most recent interaction outcome with each type of student available. This means that for a student of any type, the prediction will give the same weight for each school. Thus, by symmetry, $A$ can receive $u_1$ and $u_2$, each with probability $1/2$ at the current step. Moreover, the expected future utility remains $U$ as the current action does not change upcoming predictions. Therefore: $U = (\frac{1}{2} \cdot u_1 + \frac{1}{2} \cdot u_2) + \beta \cdot U \Rightarrow (1 - \beta) \cdot U = \frac{u_1 + u_2}{2} \Rightarrow U = \frac{u_1 + u_2}{2} \cdot \frac{1}{1-\beta}$. □

The following lemma looks at the changes in the chance of being allocated the desired type of student when the school in question interacted worse than the other school with the undesired students. Note that, since the prediction accuracy is at least 0.5, $\frac{\theta \cdot (2\alpha - 1)}{4} \geq 0$; thus, Lemma 2 implies that the

probability of being allocated an unwanted student decreases after a school chooses a worse outcome than the other school for such a student.

**Lemma 2** *When school $A$ chose a worse last outcome for the undesired type of student than school $B$ (i.e., $H_A < H_B$), then the probability of $A$ being allocated such a student in the next timestep is $\frac{1}{2} - \frac{\theta \cdot (2\alpha - 1)}{4}$. With maximum trust and accuracy, this probability is $\frac{1}{4}$.*

**Proof:** Let $X^t = \{D, U\}$ where $D$ is a student of a desired type (e.g., cheap), and $U$ is a student of an undesired type (e.g., expensive). Then:

$$\mathbb{P}(\mu(A) = U) = \mathbb{P}(D \text{ chosen first by RSD}) \cdot \mathbb{P}(B \succ'_D A) + \mathbb{P}(U \text{ chosen first RSD}) \cdot \mathbb{P}(A \succ'_U B)$$

$$= \frac{1}{2} \cdot \frac{1}{2} + \frac{1}{2} \cdot \left[ \mathbb{P}(U \text{ trusts } h) \cdot \mathbb{P}(A \succ^h_U B) + \mathbb{P}(U \text{ doesn't trust } h) \cdot \mathbb{P}(A \succ_U B) \right]$$

$$= \frac{1}{4} + \frac{1}{2} \cdot \left[ \theta \cdot (1 - \alpha) + (1 - \theta) \cdot \frac{1}{2} \right]$$

$$= \frac{1}{2} - \frac{\theta \cdot (2\alpha - 1)}{4},$$

where all probabilities are given that $H_A < H_B$. When both the trust and accuracy are maximum (i.e., equal to 1), then $\mathbb{P}(\mu(A) = U) = \frac{1}{4}$. □

Next, we consider the case of one desirable and one undesirable student arriving at each timestep and schools having exhibited a history that matches their strategy. Based on the previous two results, we compute the expected future discounted utility for a school, say $A$. Its value depends on the chance of being allocated the undesirable student, which, by Lemma 2, in turn, depends on how school $A$ treated undesirable students in the past. Thus, this function branches over the three possible comparative values of $\sigma_A$ and $\sigma_B$.

**Lemma 3** *Let $X^t = \{D, U\}$ where $D$ is a student of a desired type (e.g., cheap), and $U$ is a student of an undesired type (e.g., expensive). Assume the two schools interacted so far according to the strategy profile $\sigma$ and will continue to do so (i.e., $H = \sigma$). Then, the expected utility of school $A$ is:*

$$U_A(\sigma | H = \sigma) = \begin{cases} \frac{1}{1-\beta} \left[ \frac{u(D) + u^{\sigma_A}(U)}{2} + \frac{u(D) - u^{\sigma_A}(U)}{2} \cdot \frac{\theta \cdot (2\alpha - 1)}{2} \right] & \text{, when } \sigma_A < \sigma_B \\ \frac{1}{1-\beta} \cdot \frac{u(D) + u^{\sigma_A}(U)}{2} & \text{, when } \sigma_A = \sigma_B \\ \frac{1}{1-\beta} \left[ \frac{u(D) + u^{\sigma_A}(U)}{2} - \frac{u(D) - u^{\sigma_A}(U)}{2} \cdot \frac{\theta \cdot (2\alpha - 1)}{2} \right] & \text{, when } \sigma_A > \sigma_B \end{cases}$$

**Proof:** When the schools choose the same outcomes (i.e., $\sigma_A = \sigma_B$) then the expected discounted utility is given by Lemma 1.

Let us assume that school $A$ has a strategy according to which school $A$ gives worse outcomes for undesirable students, i.e. $\sigma_A < \sigma_B$. Then, the expected discounted utility of school $A$ is:

$$U_A(\sigma | H) = \mathbb{P}(\mu(A) = U) \cdot u^{\sigma_A}(U) + \mathbb{P}(\mu(A) = D) \cdot u(D) + \beta \cdot U_A(\sigma | H)$$

$$= \left[ \frac{1}{2} - \frac{\theta \cdot (2\alpha - 1)}{4} \right] \cdot u^{\sigma_A}(U) + \left[ \frac{1}{2} + \frac{\theta \cdot (2\alpha - 1)}{4} \right] \cdot u(D) + \beta \cdot U_A(\sigma | H)$$

$$= \frac{1}{1-\beta} \left[ \frac{u(D) + u^{\sigma_A}(U)}{2} + \frac{u(D) - u^{\sigma_A}(U)}{2} \cdot \frac{\theta \cdot (2\alpha - 1)}{2} \right].$$

In the above computation, we used Lemma 2 for substituting the two probabilities in the first line.

With a similar computation as above, when $\sigma_A > \sigma_B$, school $A$ has a higher chance of being allocated the undesired student, thus giving $A$ an expected discounted utility of:

$$U_A(\sigma | H) = \frac{1}{1-\beta} \left[ \frac{u(D) + u^{\sigma_A}(U)}{2} - \frac{u(D) - u^{\sigma_A}(U)}{2} \cdot \frac{\theta \cdot (2\alpha - 1)}{2} \right].$$

The result follows when combining the formulas for the three cases for the strategy profile. □

Finally, we look at differences in expected utility of one school, say $A$, under a strategy $\sigma_A$ and under the same strategy as $B$ (i.e., $\sigma_A = \sigma_B$) when $A$ was allocated an undesirable student. This

result prepares our analysis of best responses of school $A$ for a given strategy of $B$ after having the opportunity do provide such a response (i.e., being allocated an undesirable student and, thus, having to choose an outcome for their interaction).

**Lemma 4** *Let $X^t = \{D, U\}$ where $D$ is a student of a desired type (e.g., cheap), and $U$ is a student of an undesired type (e.g., expensive). If the strategy and history of school $B$ is an outcome $\sigma_B$ and $A$ was allocated an undesirable student (e.g., expensive), then the gap in expected utility between using some strategy $\sigma_A$ and using the same strategy as $B$ is:* $\frac{u^{\sigma_A}(U) - u^{\sigma_B}(U)}{2} \cdot \frac{2-\beta}{1-\beta} \pm \frac{\beta}{1-\beta} \cdot \frac{u(D) - u^{\sigma_A}(U)}{2} \cdot \frac{\theta \cdot (2\alpha - 1)}{2}$, *with a plus when $\sigma_A < \sigma_B$ and a minus when $\sigma_A > \sigma_B$.*

**Proof:**   The expected discounted utility given a strategy profile $\sigma$, a past interaction of $B$ according to $\sigma_B$, and a current matching that assigns the undesirable student to $A$ is:

$$U_A\left(\sigma | H_B = \sigma_B, \mu(A) = U\right) = u^{\sigma_A}(U) + \beta \cdot U_A\left(\sigma | H = (\sigma_A, \sigma_B)\right).$$

By substituting $U_A\left(\sigma | H = (\sigma_B, \sigma_A)\right)$ with its value from Lemma 3 we get different values depending on the relative ordering of $\sigma_A$ and $\sigma_B$. For example, when $\sigma_A = \sigma_B$ then

$$U_A\left(\sigma = (\sigma_B, \sigma_B,) | H_B = \sigma_B, \mu(A) = U\right) = u^{\sigma_B}(U) + \beta \cdot U_A\left(\sigma = (\sigma_B, \sigma_B,) | H = (\sigma_B, \sigma_B)\right)$$

$$= u^{\sigma_B}(U) + \frac{\beta}{1-\beta} \cdot \frac{u(D) + u^{\sigma_B}(U)}{2}$$

When $A$ changes its strategy to some $\sigma_A \neq \sigma_B$, then Lemma 3 gives a different value for $U_A\left(\sigma | H = (\sigma_A, \sigma_B)\right)$. For example, by making the substitution when $\sigma_A < \sigma_B$, the gap in expected utility for school $A$ between the two strategies is:

$$U_A\left(\sigma = (\sigma_A, \sigma_B,) | H_B = \sigma_B, \mu(A) = U\right) - U_A\left(\sigma = (\sigma_B, \sigma_B,) | H_B = \sigma_B, \mu(A) = U\right) =$$

$$= \frac{u^{\sigma_A}(U) - u^{\sigma_B}(U)}{2} \cdot \frac{2-\beta}{1-\beta} + \frac{\beta}{1-\beta} \cdot \frac{u(D) - u^{\sigma_A}(U)}{2} \cdot \frac{\theta \cdot (2\alpha - 1)}{2}.$$

The gap when $\sigma_A > \sigma_B$ is similar, with a difference (instead of a sum) between the two terms.   □

### 3.3   Not all Systems are Interaction-proof

After introducing the previous lemmas, we proceed to prove the results in the first subsection. As mentioned in the main text, these results are given under the assumption of perfect trust and accuracy (i.e., $\alpha = \theta = 1$). Since the previous lemmas support general values of $\alpha$ and $\theta$, we explicitly mention this assumption in the statement of the theorem.

**Theorem 1** *Assume $X^t = \{C, E\}$, schools are sufficiently forward-looking ($\beta > 2/3$), and predictions have perfect trust and accuracy (i.e., $\alpha = \theta = 1$). Then, schools have an incentive to deviate from truthfully interacting and implement adversarial interaction attacks.*

**Proof:**   Assume both schools interacted truthfully until now, i.e., $H = (1, 1)$. When school $A$ is matched with an expensive student, it has an incentive to attack at a level $l \in (0, 1]$ (i.e., unilaterally deviate to a strategy $\sigma_A = 1 - l < 1$) if and only if the gain in utility is greater than 0. By using Lemma 4 with expensive students as the undesirable ones, this is the case if and only if:

$$\frac{u^{\sigma_A}(E) - u^1(E)}{2} \cdot \frac{2-\beta}{1-\beta} + \frac{\beta}{1-\beta} \cdot \frac{u(C) - u^{\sigma_A}(E)}{2} \cdot \frac{1 \cdot (2 \cdot 1 - 1)}{2} > 0$$

$$\Leftrightarrow \frac{(1-l)/4 - 1/4}{2} \cdot \frac{2-\beta}{1-\beta} + \frac{\beta}{1-\beta} \cdot \frac{1 - (1-l)/4}{4} > 0$$

$$\Leftrightarrow \beta > \frac{2}{3} \cdot \frac{2l}{1+l}$$

Since the level of attack $l \in (0, 1]$ and $\frac{2l}{1+l}$ is increasing on this interval, thus having a maximum of 1, the inequality is always true when $\beta > 2/3$. Thus, when $A$ is sufficiently forward-looking, $A$ gains in expected discounted utility by interacting strategically.   □

**Theorem 2** *Assume predictions have perfect trust and accuracy (i.e., $\alpha = \theta = 1$). When there are only expensive students, a strategy profile where both schools interact truthfully is a Nash Equilibrium.*

**Proof:** There are two cases to consider: (1) in each round, there are at least two students to be matched, and (2) in each round, there is only one student to be matched.

In the first case (two students), since all students are of the same type, each school will be matched with a student of that type for any expressed preferences. Thus, the expected discounted utility of $A$ under strategy $\sigma_A$ when $B$ interacts truthfully is equal to:

$$U_A\left(\sigma | H_B = 1, \mu(A) = E\right) = \sum_{k=0}^{\infty} \beta^k \cdot u^{\sigma_A}(E),$$

which is maximized when $\sigma_A = 1$.

In the case of one student available each round, if school $A$ unilaterally interacts strategically, then each future student will be recommended, and thus prefer, $B$. Therefore, when $\sigma_A < 1$,

$$U_A\left(\sigma | H_B = 1, \mu(A) = E\right) = u^{\sigma_A}(E).$$

As a result, strategic interaction leads to a smaller expected discounted utility than when interacting truthfully (as the utility under truthful interaction is greater than $u^1(E)$, which in turn is greater than $u^{\sigma_A}(E)$).

Thus, $A$ is not incentivized to unilaterally change its strategy. By symmetry, neither is $B$. Thus, truthful interaction is a Nash Equilibrium[2]. $\square$

**Theorem 3** *Assume predictions have perfect trust and accuracy (i.e., $\alpha = \theta = 1$). When all students induce the same negative utility for schools and there are at least two students, a strategy profile where both schools interact truthfully is a Nash Equilibrium.*

**Proof:** This proof is analogous to the first case in the proof of Theorem 2. As in that case, a strategic interaction will only decrease the 1-step utility of the school while not affecting future matchings. $\square$

**Theorem 4** *Assume predictions have perfect trust and accuracy (i.e., $\alpha = \theta = 1$). If there is only one student giving schools a negative utility for any outcome and schools are sufficiently forward-looking, then schools have an incentive to deviate from truthfully interacting and implement adversarial interaction attacks.*

**Proof:** The idea of the proof is similar to the second case in the proof of Theorem 2. As in that case, the expected utility when interacting strategically ($\sigma_A < 1$) with a student giving a negative utility, say of type $N$, is

$$U_A\left(\sigma = (\sigma_A, 1) | H_B = 1, \mu(A) = N\right) = u^{\sigma_A}(N).$$

If school $A$ continues to interact truthfully, then, by Lemma 1, its expected discounted utility is

$$U_A\left(\sigma = (1, 1) | H_B = 1, \mu(A) = N\right) = u^1(N) + \frac{\beta}{1 - \beta} \cdot \frac{u^1(N)}{2}.$$

Thus, $A$ has an incentive to interact strategically whenever it gains utility, i.e. when:

$$u^{\sigma_A}(N) - u^1(N) - \frac{\beta}{1 - \beta} \cdot \frac{u^1(N)}{2} > 0$$
$$\Leftrightarrow 2(1 - \beta) \cdot (u^{\sigma_A}(N) - u^1(N)) - \beta u^1(N) > 0$$
$$\Leftrightarrow \beta(u^1(N) - 2u^{\sigma_A}(N)) > 2(u^1(N) - u^{\sigma_A}(N))$$

Since the one-step utility under truthful interaction is greater than under strategic interaction, and utilities are negative, then $u^1(N) - 2u^{\sigma_A}(N) > 0$. As a result, the above inequality is equivalent to

---

[2]We note the same proof holds for a more general scenario, when (a) all students are of the same type, say $T$, and (b) $u^o(T)$ is non-negative with a maximum at $o = 1$. In particular, the statement holds when there are only cheap students.

$\beta > \frac{2(u^1(N) - u^{\sigma_A}(N))}{u^1(N) - 2u^{\sigma_A}(N)}$. But $0 < 2(u^1(N) - u^{\sigma_A}(N)) < u^1(N) - 2u^{\sigma_A}(N)$, and thus the fraction giving the lower bound for $\beta$ is in $(0, 1)$. Since the space of $\sigma_A$ is discrete, for any $u^1(N) < 0$, there always exists some value of $\beta$ such that $\beta > \frac{2(u^1(N) - u^{\sigma_A}(N))}{u^1(N) - 2u^{\sigma_A}(N)}$ for all $\sigma_A$. Therefore, for such $\beta$, school $A$ gains in expected utility by using strategic interactions. $\qquad\square$

### 3.4 Once a school implements an attack, the other school has an incentive to respond with more severe attacks

Next, we use the lemmas to investigate the best responses of the schools after one implements a strategic interaction attack. We continue to assume perfect trust and accuracy (i.e., $\alpha = \theta = 1$).

**Theorem 5** *Assume predictions have perfect trust and accuracy (i.e., $\alpha = \theta = 1$). When $X^t = \{C, E\}$ and schools are sufficiently forward-looking ($\beta > 4/5$), the best response of school $A$ for a strategy $B$ of attacking at a level $l$ is to attack at a level $\min\{1, l + 1/L\}$.*

**Proof:** First, we show that $A$ gains more from a strategy $\sigma_A < \sigma_B$ than from $\sigma_B$ (when $\beta > 4/5$). This is the case if and only if the gap in the expected discounted utility of $A$ between using a strategy $\sigma_A$ and $\sigma_B$ is greater than 0. Using Lemma 4 this is equivalent to:

$$\frac{u^{\sigma_A}(E) - u^{\sigma_B}(E)}{2} \cdot \frac{2 - \beta}{1 - \beta} + \frac{\beta}{1 - \beta} \cdot \frac{u(C) - u^{\sigma_A}(E)}{2} \cdot \frac{1}{2} > 0$$

$$\Leftrightarrow \frac{\sigma_A/4 - \sigma_B/4}{2} \cdot \frac{2 - \beta}{1 - \beta} + \frac{\beta}{1 - \beta} \cdot \frac{1 - \sigma_A/4}{4} > 0$$

$$\Leftrightarrow 2(\sigma_A - \sigma_B) \cdot (2 - \beta) + \beta \cdot (4 - \sigma_A) > 0$$

$$\Leftrightarrow \beta \cdot (4 - 3\sigma_A + 2\sigma_B) > 4(\sigma_B - \sigma_A)$$

$$\Leftrightarrow \beta > \frac{4(\sigma_B - \sigma_A)}{4 - 3\sigma_A + 2\sigma_B},$$

where the last inference is possible since $0 \leq \sigma_A < \sigma_B \leq 1$, and thus $4 - 3\sigma_A + 2\sigma_B > 0$. The maximum of the fraction on the right-hand side is achieved when $\sigma_A = 0$ and $\sigma_B = 1$; therefore, the statement is always true when $\beta > 2/3$ (and thus also when $\beta > 4/5$).

Second, we note that Lemma 4 implies that the gain in expected utility by using $\sigma_A < \sigma_B$ instead of $\sigma_B$ is a monotonously increasing function in $\sigma_A$. Thus, the highest expected utility for $A$ under $\sigma_A < \sigma_B$ is for the highest possible value of $\sigma_A$. Given our space of actions, this is for $\sigma_A = \sigma_B - 1/L$ (i.e., an attack at a level $l + 1/L$).

Third, we show that $A$ always gains more by using strategy $\sigma_B$ than by using strategy $\sigma_A > \sigma_B$. Note that this step is only necessary when $\sigma_B < 1$. Using Lemma 4 and a similar reasoning as for the first part of this proof, we get that $A$ benefits more from strategy $\sigma_B < 1$ than from a strategy $\sigma_A > \sigma_B$ if and only if

$$\beta > \frac{4(\sigma_A - \sigma_B)}{4 + \sigma_A - 2\sigma_B},$$

which is always true when $\beta > 4/5$.

By the three parts above, when $\beta > 4/5$, the best response of $A$ when $B$ attacks at a level $l$ is to attack at a level $\min\{1, l + 1/L\}$. $\qquad\square$

**Corollary 1** *Assume predictions have perfect trust and accuracy (i.e., $\alpha = \theta = 1$). When $X^t = \{C, E\}$ and schools are sufficiently forward-looking, the strategy profile where both schools use adversarial interaction attacks at the highest level (i.e., $l = 1$) is a Nash Equilibrium. However, this is Pareto dominated by the strategy profile where all schools interact truthfully.*

**Proof:** The first part of this Corollary is a direct consequence of Theorem 5. More precisely, Theorem 5 says that when both schools attack at the highest level (i.e., $l = 1$) then each school plays a best response to the strategy of the other. This means that $\sigma = (0, 0)$ (i.e., a 0 outcome for expensive students) is a Nash Equilibrium with respect to the considered set of strategies.

For the second part, we use Lemma 1. The expected utility of each of the schools when interacting truthfully (i.e., under $\sigma = (1, 1)$) is $\frac{1}{1-\beta} \cdot \frac{1 + 1/4}{2}$ while when interacting strategically (i.e., under

$\sigma = (0,0)$) this utility decreases to $\frac{1}{1-\beta} \cdot \frac{1+0}{2}$. Thus, the Nash Equilibrium $\sigma = (0,0)$ is Pareto dominated by the strategy profile where all schools interact truthfully. □

## 3.5 Increases in trust in and accuracy of recommendations lead to higher discounted expected utilities under adversarial interaction attacks

The final theorems serve to help us understand what happens as the trust in and accuracy of predictions improves. Each of their proofs is based on Lemma 4.

**Theorem 6** *When $X^t = \{C, E\}$, the expected utility of a school using an adversarial interaction attack is a continuous function that increases monotonously in both accuracy ($\alpha$) and trust ($\theta$).*

**Proof:** We base this proof on Lemma 4. Assume school $A$ implements an adversarial interaction attack. Thus, $A$ chooses outcomes $\sigma_A < \sigma_B$ for expensive students. By Lemma 4, the gap in expected utility of $A$ when interacting according to $\sigma_A$ instead of $\sigma_B$ is $C_1 + C_2 \cdot \frac{\theta \cdot (2\alpha - 1)}{2}$, where $C_1, C_2$ are constants with respect to $\alpha$ and $\theta$, and $C_2 > 0$ (since $u(C) - u^{\sigma_A}(E) > 0$ and $\beta \in [0, 1)$). Thus, this gap is continuous and monotonously increasing in both $\theta \geq 0$ and $\alpha \geq 1/2$.

If $B$ interacts truthfully, the above means that the gain of $A$ by interacting strategically (as opposed to truthfully) is a continuous function that increases monotonously in both $\theta \geq 0$ and $\alpha \geq 1/2$. Otherwise, $\sigma_B < 1$. By using again Lemma 4, the gap in expected utility of $A$ when interacting truthfully and according to $\sigma_B$ is $C_1' - C_2' \cdot \frac{\theta \cdot (2\alpha - 1)}{2}$, where $C_1', C_2'$ are again constants with respect to $\alpha$ and $\theta$, and $C_2' > 0$. By subtracting the two, we obtain that $A$ gains $(C_1 - C_1') + (C_2 + C_2') \cdot \frac{\theta \cdot (2\alpha - 1)}{2}$. Since both $C_2$ and $C_2'$ are positive the result follows also in the case where $\sigma_B < 1$. □

**Theorem 7** *When $X^t = \{C, E\}$, the truthful strategy profile is a Nash Equilibrium under predictions with 50% accuracy ($\alpha = 0.5$) or null levels of trust ($\theta = 0$).*

**Proof:** Assume $\alpha = 0.5$ or $\theta = 0$, which implies $\frac{\theta \cdot (2\alpha - 1)}{2} = 0$. We consider one school, say $A$. By the proof of Theorem 6, when $B$ has and will interact according to $\sigma_B$ the gap in the expected discounted utility of $A$ under truthful and strategic interaction is either $C_1$ (when $\sigma_B = 1$) or $C_1 - C_1'$ (otherwise). By substituting the values for $C_1$ and $C_1'$ from Lemma 4, this gap is $\frac{u^{\sigma_A}(E) - u^1(E)}{2} \cdot \frac{2 - \beta}{1 - \beta}$ for any $\sigma_B$. Since strategic interaction lowers the one-step utility (i.e., $u^{\sigma_A}(E) < u^1(E)$), truthful interacting is a best response for school $A$ for any strategy of $B$ (i.e., truthful interaction is, in fact, a dominant strategy). In particular, this implies that the strategy profile where both schools interact truthfully is a Nash Equilibrium. □

# 4 Experiment

In the main text, we used theoretical analysis to prove that parties might have an incentive to interact strategically. An alternative way to operationalize the framework is by creating and simulating an agent-based model (ABM) for school choice. The parameters used in the experiment are summarized in Table 1, and the code is available on GitHub [3].

## 4.1 The Model

*Attributes for students and schools.* Each student and school has an associated vector of attributes. Building on the model proposed by Chen and Sönmez [5], the dimensions of these attributes correspond to different evaluation criteria (e.g., level in Math/English, or Science/Arts). The values for the attributes are integer ratings on a scale from 0 to 5. For students, the attributes reflect their current level of knowledge, while for schools, it shows their potential to help students. Throughout the experiment, we assume all schools have a maximum potential level; this corresponds to an idealized scenario where schools can help students achieve the best possible outcome. For students, the attributes are obtained by taking a normally distributed random number and rounding it to the

---

[3]GitHub link: https://github.com/StefaniaI/Predictions-MM.

| Parameters | Values Taken |
|---|---|
| **Varied in Experiment** | |
| # of schools | 2, **10** |
| Attributes of students $\sim \mathcal{N}(\mu, \sigma)$ | $\mu \in \{\mathbf{1}, 3.2\}$,, $\sigma \in \{0.65, 1.5, \mathbf{3}\}$ |
| Competition (# students per place) | 4, **1**, 1/4 |
| Sign of utility per student | **positive**, negative |
| Cost per improvement, $\alpha$ | 0.5, **0.95** |
| Matching mechanism | SD, **RSD**, Boston, DA |
| Level of prediction noise | **0.01%**, 1%, 10%, 30% |
| # of past rounds used for training | 1, **3**, 5 |
| # of neighbours (k) for KNN | **1**, 3, 5 |
| Level of trust in recommendations | 0.5, **1** |
| Level of adversarial attack (in %) | 0, **4**, 25, 50, 75, 100 |
| **Fixed in Experiment** | |
| # of evaluation-relevant attributes | 1 |
| Evaluation scale for each attribute | 0 - 5 |
| # of students | 20 × (# of schools) |
| School capacity | 20 ÷ (competition) |
| Noise of student observations | 1% |
| # of rounds capturing the utility | 100 |
| # of random seeds per run | 20 |

Table 1: Tabular description of model parameters (Left) and the values taken in the experiment (Right). The values in bold are the default ones, i.e. the varied parameters are kept at their value shown in bold unless specifically said otherwise.

nearest integer on the rating scale. Last, for the presented experiments, we use only one dimension for the attribute vector (e.g., the GPA). [4]

*Outcomes.* When a student is assigned to a school, the school decides on the outcome of the interaction. The available outcomes depend on students' and schools' attributes. More precisely, on each attribute, the school can either:

- Put in the *standard effort*: the student will exit the school with a value equal to the minimum between their entry knowledge and the potential of the school;

- *Help the student improve*. If the potential of the school is higher than the entry knowledge of the student, the school can choose how much it will help the student improve; the maximum help is the difference between the attributes of the school and of the student.

Using the notation introduced before, if a school $y$, with attribute $a_y$, is matched with a student $x$ with attribute $a_x$, then $\mathcal{O}_y(x) = \{a | a \in [\min(a_x, a_y), \max(a_x, a_y)]\}$. For example, if a school with a level of 5 interacts with a student with knowledge 3, according to our model, the outcome of the interaction is a student-level of $a \in [3, 5]$. Moreover, the help given by the school to the student is equal to $a - 3$.

*Value of outcomes.* The value of an outcome is equal to the outcome, i.e., the attribute of the student when exiting the school. In other words, $v_y^o(x) = o$.

*Cost of outcomes.* Depending on the outcome, the school encounters a cost. We assume the cost is the total level of help discounted by a factor of $\alpha$, i.e. $c_y^o(x) = \alpha \cdot (o - a_x)$.

*Utility of outcomes.* The utility of the school is the value minus the cost, i.e., $u_y^o(x) = o - \alpha \cdot (o - a_x)$. Note that if $\alpha < 1$, helping the student as much as possible always gives the highest one-step utility for the school and this utility is always positive. To account for the case when there are high integration costs, so the returning side prefers not to receive agents (e.g., for refugee assignment or underperforming student re-assignment), we also allow for negative utility. This is achieved by subtracting the maximum student rating (i.e., the constant 5) from the utility; we test both scenarios (see Table 1).

---

[4]In the initial phases of the experiments, we used multiple attribute dimensions. However, once all schools have the maximum potential to help students, the only effect of having multiple attributes is that of changing the mean and standard deviation of the outcome-value distribution. Therefore, we only use one attribute but vary the mean and standard deviation of their distribution.

## 4.2 Preference Formation and Strategies

*Strategies of schools.* We tested two strategies for the schools. First, we have the *truthful* one, in which schools help students as much as they can. Second, we have a *strategic* interaction. Here, schools distinguish between two categories of students: *cheap* (i.e., students that require less than a threshold, $t_y$, of help from the school $y$), and *expensive* (i.e., students that require more help than that threshold). Under strategic interaction, the school treats the students differently depending on the group they are in: If a student is considered cheap, then the school helps the student as much as possible; otherwise, the school only helps the student to achieve less than the best possible outcome. More precisely, if a student is considered expensive, the school will help the student achieve the best possible outcome minus $l\%$ of the maximum rating. We refer to $l$ as the *level of adversarial attack*. Going back to the previous example, for a school of level 5, a student of knowledge 3 considered expensive by the school, and a level of attack of $4\%$, the school will only help the student reach an outcome of $4.8$, instead of 5.

*Threshold Cheap-Expensive in Strategies.* The choice of value for the threshold makes an important difference on the strategy. For example, if $t_y$ is 5 then all students are considered cheap and we obtain the truthful behavior. Differently, if $t_y$ is negative, then all students are considered expensive and the school does not help any student achieve their maximum level (for non-zero levels of attack). The best threshold depends on the particularities of the system – especially on the competition and on the sign of the utility per student. Hence, we set the default threshold depending on the parameters. More precisely, we make this choice based on three factors:

- First, schools decide on how many students they want to be matched with. When the utility per student is positive, the school wants as many students as possible, i.e., the minimum between the number of students and the school's capacity. Otherwise, the school only wants students that exceed the capacities of the other schools, subject to their own capacity constraints.

- Second, the schools take into account that they will face competition for the students they want. Therefore, when interacting strategically, they aim to get the cheapest $50\%$ of the number of students wanted times the number of schools.

- Third, the schools look at the distribution of attributes and infer the expected number of upcoming students for each level of help. The threshold is obtained using this distribution and the number of students the school aims to get. More precisely, the threshold $t_y$ is the minimum $t$ such that there are at least that number of students considered cheap (i.e., requiring at most $t$ help to achieve the maximum outcome).

We illustrate this by an example. Let us assume utilities are negative and there are 3 schools and 40 students. If the capacity of each school is 80 then schools desire 0 students; differently, if the capacity is 15 then schools desire 10 students each. When the capacity is lower there is competition; cumulatively, schools desire the best 30 students, so, each school will aim to get the best $50\%$ of these students, i.e., the cheapest 15 students. The remaining ones are targeted by the attack. The threshold is chosen based on the distribution of attributes for the level of knowledge. More precisely, the school $y$ chooses $t_y$ such that the expected number of students coming in the following year and requiring at most $t_y$ help is 15.

*Prediction Algorithm.* Similar to previous work, we use k-Nearest Neighbour (KNN) to predict academic performance [9, 3]. More precisely, based on the history of interactions in the most recent rounds (years), the algorithm finds the closest $k$ past students in terms of entry attributes assigned to each school, averages their outcome, and uses this average as the prediction for the current student. The prediction is prone to some observation noise which reduces its accuracy. To model it, the predicted outcome varies by $\pm\mathcal{U}(5 \cdot p/100)$. We alter the level of noise (see Table 1).

*Student preferences.* Students form their preferences based on the predictions and their own observations. We assume the observation of the student is given by the prestige of the school, i.e. the average evaluation score of the outcomes of students who attended the school in the previous year. For each school, the student weights the school as the linear combination between the predicted outcome and their own observation. The importance given to predictions (i.e., *level of trust*) is varied, as shown in Table 1.

*Matching mechanisms.* We implemented 3 commonly used school choice matching mechanisms, namely Serial Dictatorship (SD), Boston, and Deferred Acceptance (DA)[5]. In practice, these mechanisms differ depending on the ordering of students. For the serial dictatorship mechanism, the students could either be ordered at random (RSD) or, as in, e.g., Mexico City [8], by the exam-measured entry-level (SD). Similarly, for DA and Boston, the preference of schools could either be given by a random order, or by the true preference of schools. When deployed in the school choice setting, the preferences of schools are usually given by a lottery with some priority ordering (e.g., students with siblings at the same school have a higher priority) [1]. Unless explicitly mentioned otherwise, DA and Boston refer to their respective versions using lotteries.

## 4.3 Outcomes Measured in the Experiment

We run the experiment to capture four key aspects regarding adversarial interaction attacks.

*Incentives to attack.* To see whether schools have an incentive to interact strategically, we compare the utility of one school [6], $A$, under two scenarios: (a) when all schools, including $A$, interact truthfully and (b) when $A$ interacts strategically while all other schools interact truthfully. If the utility of $A$ increases when it unilaterally interacts strategically, then $A$ has an incentive to deviate from interacting truthfully and adopt such a strategy.

*The effect of accuracy of and trust in predictions.* The impact of predictions depends on two key factors, namely their accuracy and trust. Consequently, system designers usually try to improve on those two metrics [13, 7]. We analyze how such improvements impact the benefits from adversarial interaction attacks.

*Best responses of other schools.* We use a game-theoretic framework to understand what the response of other schools will be once a school interacts strategically. To do so, we consider a game with schools as the agents. For computational tractability, we restrict the set of actions to adversarial interaction attacks of levels $0\%, 25\%, 50\%, 75\%$, and $100\%$. The utility for each action is the expected long-term utility according to the simulation. With respect to this set of actions, each school has a best response (i.e., an action that gives it the highest utility) given the actions of others. Initially, all schools have an attack level of $0\%$. Then, they take turns finding their best response to the current profile (i.e., they find an action that increases their expected utility the most and significantly). If this process terminates, then the final action profile is a *Nash Equilibrium* (i.e., a choice of action when all schools are playing a best response to the other's actions and, thus, none of them can change their level of attack to achieve a higher expected utility).

*Student Welfare.* Interacting strategically also affects the welfare of students. Last, we report how this metric, as measured by the average outcome of students, changes as schools adapt their strategies.

## 4.4 Computing Infrastructure

The experiment was designed for Python 3.8.10. For successfully running the simulations one needs the following Python libraries: *numpy*, *pandas*, *scipy*, *csv*, and *copy*. In addition, the visualization functions require *matplotlib* and *seaborn*. We run the simulation on a machine with the following specifications:

- OS: Ubuntu 18.04.5 LTS

- RAM: 32GB

- CPU: Intel® Core™ i7-6700 3.40GHz × 8 cores

- GPU: GeForce GTX 1060 6GB/PCIe/SSE2

---

[5]See Section 2.2 in the Supplementary Matherial for a description of each mechanism

[6]The values (e.g., utilities) obtained via simulations are in fact, the mean of the values obtained by running the simulation setup several times but with different random seeds. When comparing two values we say that their difference is *significant* if they are more than the sum of their standard deviations apart.

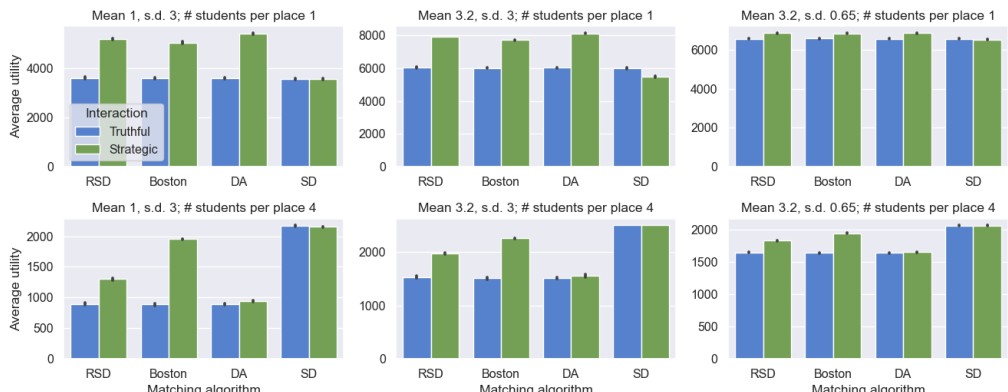

Figure 1: Figures showing the utility of one school when it interacts truthfully and strategically; all other schools interact truthfully. We also vary different aspects of the system: the matching mechanism, the distribution of student attributes, and the level of competition.

## 5 Simulation Results

### 5.1 Not all Systems are Interaction-proof

As shown in Figure 1, schools usually have an incentive to interact strategically under matching mechanisms that use lotteries to differentiate between students. [7] There are, however, important differences depending on other model parameters. First, depending on the competition level, different mechanisms are more susceptible to interaction attacks. If the number of students is equal to the number of available places, then a school behaving strategically increases its utility the most under DA, namely by a little over $50\%$. In contrast, if there is a competition of $4$ students per available place at the schools, then Boston is the mechanism that induces the highest increase in utility: in this scenario, the school more than doubles its utility by interacting strategically. When the preferences of schools are given by the entry knowledge of the student, attacks are no longer beneficial (see SD). This suggests that, when accurate and trusted recommendations are available, market designers should consider letting the returning agents express their preferences freely. Second, the distribution of the initial levels of the students affects the gains obtained through attacks. More precisely, schools gain more with the decrease of mean and the increase of variance. We included here the figure for a mean of $3.2$ and a standard deviation of $0.65$ as this choice of parameters induces a similar distribution to that of the SAT scores of students[8]. However, note that here we assume a one-to-one correspondence between the measured knowledge of a student and the respective value gained by the school. This is not necessarily the case; e.g., a school might not value differently students with scores (out of 800) of 760 and 800, respectively, but might find a larger difference between students with scores of 680 and 720, respectively.

When there are more available spots in schools than students to fill them, schools do not have an incentive to interact strategically when utilities are positive, but do when utilities are negative (see Figure 2). For positive utilities, schools compete for students; therefore, when interacting strategically, a school sets a threshold of 0 and, thus, aims to give the maximum help to all students. Consequently, both the strategic and the truthful behaviour produce the same average utility. This is different when utilities are negative as, then, schools prefer not to have students assigned. In other words, the low number of students per available place leads schools to consider all students expensive, and, thus, to interact strategically with all students assigned to them. This behavior triggers the RS to predict a lower outcome at the school using the attack for all students. Consequently, students usually rank the school last and, therefore, do not get assigned to it. So, the strategy triggers a significant increase in the utility of the school.

---

[7]For Figure 1, the student competition level is at least $1$; in this case, setups with positive and negative utility are qualitatively equivalent, so we only included plots for the positive utility setups.

[8]https://reports.collegeboard.org/pdf/2020-total-group-sat-suite-assessments-annual-report.pdf

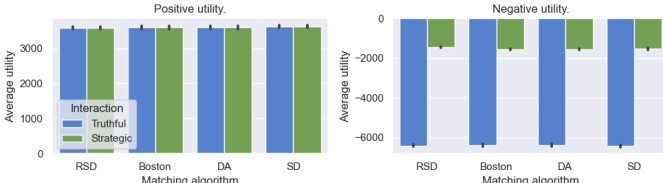
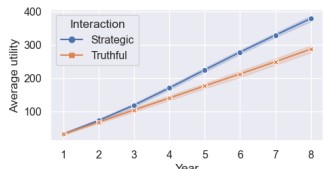

Figure 2: Figures showing the utility of one school when it interacts truthfully and strategically; all other schools interact truthfully. The student-competition level is $1/4$. The figure shows both the case of positive and negative utility.

Figure 3: Figure showing the utilities with and without strategic interaction depending on the number of years elapsed.

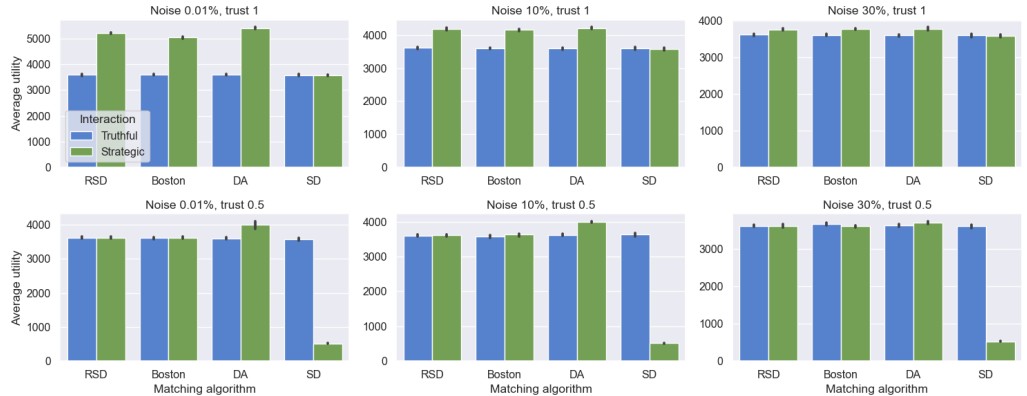

Figure 4: Figures showing the utility of one school when it interacts truthfully and strategically; all other schools interact truthfully. We vary the matching mechanism, trust in the RS, accuracy of the RS.

Although previously we reported on the utilities of schools over a period of $100$ years, we want to point out that schools start having an incentive to interact strategically much faster. In fact, a school has a significant boost in utility by implementing the attack starting from its third year of using it. The cumulative utility of a school with and without interacting strategically over a period of at most $8$ years is shown in Figure 3. [9]

## 5.2 The Effect of Higher Accuracy of and Trust in the RS

Figure 4 shows that, as RSs become more accurate and trusted, schools gain more by interacting strategically (under mechanisms using lotteries for the preferences of schools) and lose less (under SD). In fact, when there is a $30\%$ noise level in the predictions and the weight of recommendations is only half, there is no significant utility gain obtained by interacting strategically. This suggests that in such scenarios schools will not have an incentive to implement adversarial interaction attacks. This adds to the existing discussion on why aiming for accuracy alone when designing a RS could be harmful [13] by presenting yet another competing objective.

## 5.3 Responses of Other Schools to Adversarial Interaction Attacks

We find that a school that competes against other schools which are using adversarial interaction attacks also benefits from implementing such an attack itself. According to the best response analysis,

---

[9]In Figure 3, all parameters are kept at their default values according to Table 1. However, with the increase of the attack level and the decrease of the cost of schools for each unit of help given to a student, $\alpha$, the number of years required to see a significant utility gain by attacking increases. For example, when $\alpha = 0.5$, a $25\%$-level attack only produces a significant gain starting from the fifth year, while a full attack only produces a significant gain starting from the 45th year. Moreover, up to the third and, respectively, 16th year schools suffer a significant loss in utility by attacking.

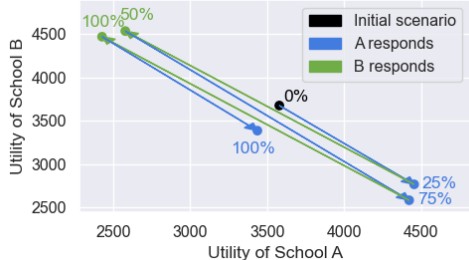

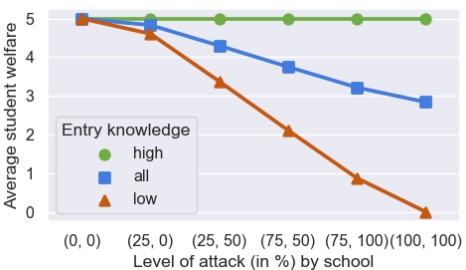

(a) The change in the utility of schools as they adapt their strategies. Initially, both schools interact truthfully. Schools in turn change their strategy to best respond. Arrows indicate the direction of change, and annotations – the new level of attack.

(b) Figure showing the average welfare of students depending on the level of attacks of schools. Students are grouped based on their entry level of knowledge: all (i.e. any level between 0 and 5), low (i.e. level of 0), high (i.e. level between 1 and 4).

Figure 5: Figures capturing (a) the utility of schools and (b) the welfare of students by entry level of knowledge when 2 schools are in turn best-responding to the strategy of the other.

each school in turn adopts an attack of a higher level than the one previously used by the other school. The results for simulating a two-school scenario are shown in Figure 5a. First, both schools interact truthfully. Next, School A best responds by attacking at a level of $25\%$; this change in action increases the utility of School A, and decreases the utility of School B. Similarly, in the next round, School B increases its utility at the expense of the utility of School A by attacking at a higher, $50\%$, level. This continues until both schools attack at the maximum level (i.e. they do not help students considered expensive at all, thus leaving them at their entry level). From this point, neither of the schools benefits from unilaterally changing its action; therefore, this is a Nash Equilibrium. This equilibrium is, however, undesirable for both schools, as it is Pareto dominated by the initial choice of actions, i.e. if both schools interact truthfully, then both of their utilities are higher than when the schools attack at the maximum level.

## 5.4 The Effect of Strategic Interaction on Student Welfare

Figure 5b shows the average outcome of students depending on the attack levels of each school. As expected, adversarial interaction attacks decrease the welfare of students (as measured by their average outcome) proportional to the attack level of schools. In particular, when attacking at a full level, the average outcome of students drops by around $43.2\%$. Moreover, this increases the disparity between students, as not all are affected equally by the attacks. All loss in welfare is supported by the students that are targeted by the attacks, i.e. the ones that have a low initial level of knowledge and are thus considered expensive by the schools. For the chosen distribution of attributes, the targeted students are the ones having a low (i.e. 0) entry level of knowledge. Altogether, Figure 5 shows that the equilibrium with respect to the considered set of actions is non-optimal for both schools and targeted students.

# 6 Future Work

## 6.1 Extension for Experiment - Multiple Dimensions for Attributes

We originally designed the model to account for multiple attribute dimensions (e.g., Math and English scores instead of GPA alone). To do so, we interpret each student attribute as their level of knowledge/expertise in some domain and each school attribute as its potential to help the student improve in that domain. The set of outcomes from an interaction was obtained with the same rules applied component-wise. That is, on each component, a student can achieve an outcome between their original level and the potential of the school. The value of an outcome was the sum of its components.

To give a formal definition of this, assume a school $y$, with attributes $a_y$, is matched with a student $x$ with attributes $a_x$. Then the set of outcomes is $\mathcal{O}_y(x) = \{a | a_i \in$

$[\min((a_x)_i, (a_y)_i), \max((a_x)_i, (a_y)_i)]\}$. As an example let us assume that a school with a level of 3 for both Math and English interacts with a student with levels 5 and 1 respectively. Then, the school has attributes $(3, 3)$, while the student has attributes $(5, 1)$. As a result, according to our model, the outcome of the interaction could be a student level of $(3, x)$, where $x \in [1, 3]$. Moreover, the help given by the school is equal to $x - 1$ and the value of the outcome is $3 + x$.

Having multiple dimensions of attributes with potentially different distributions of values for each dimension and different functions for computing the values per outcome are all interesting extensions of our work. As a starting point, the publicly available code [10] accounts for the extension mentioned above. In addition, it also allows for measuring the value of an outcome as the minimum of the values per component.

## 6.2 Interventions

An important question that we leave for future work is how to design an interaction-proof system. Our initial analysis suggests the gains from attacks largely depend on the specifics of the market. Thus, efficient interventions will likely depend on the application domain and might require input from domain experts to assess their feasibility. Ideas include cutting the feedback loop by ignoring some data, maintaining some noise in predictions (see Section 5.3, although finding the optimal $\alpha$ would be difficult in practice), giving more decision-power to returning agents (e.g., by allowing them to fully express their preferences, see the simulation results), or using a different matching mechanism. An alternative solution could involve legislation: while detecting and punishing adversarial interactions is likely difficult to impossible, there could be a reallocation of funds to counteract the undesirability of matches. Doing so could make utilities of non-returning agents positive and uniform (thus, by Theorem 2, making the system interaction-proof). However, these remain just some theory-inspired ideas, and we believe a thorough consideration of the application domain would be needed before implementing or adapting any of them. In order to avoid giving a hasty suggestion, we concentrated in this work on underlining the main causes of problems, kept the discussion on potential solutions at a high level, and largely reserved this analysis for future work.

## 6.3 Additional Extensions

There are several additional extensions for our work. First, we only investigated one type of ML-algorithm; the efficiency of attacks under different algorithms is still unknown. Second, a decisive factor in using adversarial attacks is to be undetectable, which we did not investigate. Introducing good auditing procedures might make some types of adversarial interaction attacks unfeasible. Third, to test the attacks, we used theoretical analysis and simulations with simple models. While this was useful for isolating the effects of different parameters and discerning their effects, more realistic models, potentially extending to other application domains, will help better understand this type of strategic behavior.