# OpenReview forum: "Strategic Behavior in Two-sided Matching Markets with Prediction-enhanced Preference-formation"
_NeurIPS.cc/2023/Conference — NeurIPS 2023 poster_

### Official Review · Reviewer_UhQP · 2023-07-05

**Soundness:** 3 good
**Presentation:** 3 good
**Contribution:** 2 fair
**Rating:** 6
**Confidence:** 3

**Summary:**

This paper studies the matching market with returning agents and
proposes an important strategic behavior that returning agents can
attack future predictions by distort short-term interactions. The
authors formulate the systems as (repeated) three phases and study a
simplified setting to derive informative conclusions on the impact of
this attack behavior.

**Strengths:**


  -Novel and meaningful topic of strategic behavior in matching market
  with returning agents.

  -The description of setting and analysis of phenomenon is clear and
  easy for readers to follow.


**Weaknesses:**

  A natural question is whether these conclusions can be extended to
  larger matching markets. The setting used for inducing main
  conclusions of this paper seems too simplified.

**Questions:**

  In line 275-276, "the school always chooses outcome
  1-l", I am confused about how can the school choose
  the outcome of interaction here.

**Limitations:**

Yes.

---

> ### Author Rebuttal · Authors · 2023-08-08
>
> We thank Reviewer UhQP for their feedback and question. We are grateful both for their appreciation of the novelty and importance of the topic and for their concern about the simplicity of the model.
>
> Regarding the question, we apologize for the lack of clarity surrounding the link between adversarial interaction attacks and outcomes. In the interaction phase, the school interacts with the allocated student. During this interaction, the school has some choice of action (e.g., provide integration services or after-school preparation for the student transferring from a low-performing school). In practice, the student also has a choice to react to these opportunities (e.g., attend and pay attention to these services). However, as mentioned in lines 161-165, since the student is not returning, we assume they will make the most out of the provided opportunities. Thus each action of the school will have a direct connection with an (expected) outcome of the interaction. For example, the school could know, based on past experience, that by providing a fraction of (1-l) of the total hours of after-school training, the student (is expected to) get a final SAT score equal to a fraction (1-l) of the maximum one. As such, each action of the school during the interaction with the student corresponds to an outcome of their interaction. We tried to capture this intuition in the example included in lines 171-176. We will clarify this in the final version.
>
> The level of control a returning agent has on the interaction outcome varies with the application domain. For example, in a school choice setting, there is still some variability in the performance of the student. That is, based on the performance of the student in mock exams and/or the performance of similar past students, the school can anticipate the final result of the current student, but there will still be some variability. However, in a refugee assignment problem, if a location does not provide (good) interview opportunities for one refugee in the first 90 days, the respective refugee does not have any chance of getting a (good) job. In short, in a general scenario, the returning agent chooses their action (i.e., how much support they provide to a given non-returning agent), which maps to an expected outcome (i.e., the success of the interaction, which is estimated based on past experiences).
>
> We would also want to add a few words regarding the simplicity of the model. As mentioned in the reply for reviewer Zyns, throughout the modeling phase, we tried to follow the model-simplicity principle introduced by Robert Axelrod (1997) and used in the next decades. More precisely, our goal was to keep the model minimal in order to highlight the source of vulnerability. This also allowed us to create a general framework and , thus, show the feedback loop could be problematic in multiple settings.
>
> That being said, we are in agreement with Reviewer UhQP that besides its advantages, the simplicity of the model is also a limitation. In particular, we underline in lines 394-395 that a central direction for future work would be to consider more realistic (and in particular larger) models. As a first step, we used the appendix to extend the model (e.g., by considering higher numbers of agents, alternative matching mechanisms, and k-NN as a more realistic recommender system) and perform simulations. This additional analysis largely confirmed our theoretical results. Most important, when there are few non-returning agents compared to available places and the utility of returning agents is always negative, adversarial interaction attacks were effective for all considered matching mechanisms. While this analysis does not exclude the need for larger and more realistic settings in future work, we hope it provides some evidence that such attacks are worth investigating within various settings and application domains.

---

> > ### Comment · Reviewer_UhQP · 2023-08-15
> >
> > Thanks for the response. I have read the rebuttal and remain my score.

---

### Official Review · Reviewer_EhdA · 2023-07-06

**Soundness:** 3 good
**Presentation:** 4 excellent
**Contribution:** 3 good
**Rating:** 7
**Confidence:** 4

**Summary:**

This paper is looking at problems that arise in two sided matching markets, where preferences of the agents are being informed by various prediction mechanisms.  In particular, the paper makes an argument that the existence of a predictive model used by agents to inform their preference has interesting strategic repercussions. They argue that when thinking about market design, predictive models can not be ignored or analysed independently.



**Strengths:**

I liked this paper. It was well written and  the theoretical analysis appears to be solid. That said, I think its significance comes in clearly highlighting the arise of potentially unforeseen strategic behavior in markets due to the use of predictive models that shape users’ preferences. The argument made in the paper that there are interesting interplays between ML models and market mechanisms is an important warning to market designers AND opens up interesting theoretical questions in the mechanism design/market design space.



**Weaknesses:**

-	The theoretical analysis was focussed on the random serial dictatorship mechanism. I understand why the authors might have done so as it simplifies the analysis significantly, but theoretical results on other matching mechanisms (e.g. DA) would have also been interesting.

**Questions:**

-	Are there qualitatively similar findings if DA was used instead? What would be the complications with analyzing DA over RSD? I looked at the results in the simulation and it seemed like DA did not always follow identical trends as RSD and was curious as to its behaviour.

-	A key driver of the results seems to be the feedback loop between the prediction model and the market.  Would simply cutting this feedback loop be enough (I will admit I am not sure how this would be done without ignoring data) or are you suggesting that an entirely new way of modelling such mechanisms is needed?





**Limitations:**

The authors did a very good job at discussing the implications of their work in terms of inequity. In particular, this is a key message of the paper -- that care needs to be taken in designing and deploying market mechanisms  since data generated by them and fed into predictive models might have strategic ramifications that can exaspertate inequality.

---

> ### Author Rebuttal · Authors · 2023-08-08
>
> We thank Reviewer EhdA for their supportive feedback and insightful questions! We are especially grateful for their engagement with both the paper and supplementary material.
>
> To address the first question, as mentioned by the reviewer, DA poses additional challenges as we would need to consider more rounds (where students propose to schools) and potentially distinguish between different versions of DA. Since using DA changes the matching phase, the main difference comes in the probability of being allocated an undesirable student given a unilateral past strategic interaction (when histories are equal, we have symmetry and thus equal probabilities of outcomes). To take one example, under student-proposing DA with lotteries for the preferences of schools, there are two alternatives: (a) the two students prefer different schools (thus, each being allocated to the school they prefer), or (b) the two students prefer the same school (thus, using the lottery-based preference of the preferred school for the allocation). By interacting strategically, a school maximizes the chance of the undesirable student to prefer the other school (i.e., the beneficial subcase of a). This will complicate the proof of Lemma2 (within the appendix) but otherwise lead to the same conclusions. Since this analysis requires explaining DA and discussing the intuition behind a larger number of particular cases while providing a limited amount of additional insight, we reserved the discussion of DA for the simulation.
>
> As EhdA accurately noted, by departing from the assumptions of the minimal model, we notice differences between DA and RSD. Looking at individual rounds during matching revealed this is due to the increased importance of the lottery-based preferences of schools under DA. For instance, assume there are more students than places at schools (which produced the largest gaps DA - RSD) and two schools with one place each. Then, under RSD, two students will be selected: the first will be allocated to the school they prefer the most, and the second to the other school. Being strategic under RSD is thus effective as, when the first student is undesirable, it minimizes the chance this student will be allocated to the strategic school. On the other hand, under DA, in the first round, each student proposes to the most preferred school. In the next round, all students who applied to school B and were not tentatively accepted by B would apply to school A and vice-versa. Thus, by the third round, each school would have tentatively accepted the student they (lottery-based) prefer the most out of all but one student. Therefore, the effectiveness of a strategy used by, e.g., school A is limited to maximizing the pool of undesirable students school B receives in the first round of the DA. However, since there are still many undesirable students remaining, school A will still receive proposals from many undesirable students in subsequent rounds. Thus, besides edge cases, under DA with many undesirable students, schools receive the students they (lottery-)prefer the most.
>
> On the second question, as EhdA mentioned, cutting the feedback loop by ignoring data would be effective throughout scenarios. From the results of Section 5.3, one alternative is to maintain some noise in predictions (i.e., lower the prediction accuracy, alpha). More precisely, as discussed in lines 371-376, we want the maximum value of alpha that will introduce sufficiently much noise to make adversarial interaction attacks inefficient (under the current market parameters, e.g., beta, theta). This is, however, to some degree, a theoretical solution. While it makes sense mathematically, it is unclear how to find this optimal value of alpha in practice as many parameters (e.g., precise utility functions, how forward-looking returning agents are) are private information. Moreover, since these attacks are detrimental to social welfare, returning agents might be reluctant to share truthfully whether they would implement adversarial interaction attacks.
>
> Based on the simulation results, another solution could be giving more decision-power to returning agents. For example, by allowing them to fully express their preferences (instead of using lotteries), we might eliminate the incentive to attack when utilities are positive or all places will be filled. However, the intervention is ineffective when utilities are negative and there are fewer non-returning agents than places. We discuss this intervention in lines 401-404 but not what to do in the latter scenario (which corresponds to realistic settings, e.g., refugee assignment). As mentioned by EhdA, this might require a different mechanism design. For instance, we could use top trading cycles; start with a random allocation of, e.g., refugees to locations and redistribute refugees based on the preferences of locations. It is important to investigate, though, the resulting loss in welfare.
>
> Finally, one alternative solution could involve legislation. While detecting and punishing adversarial interactions is likely difficult to impossible, there could be a relocation of funds to counteract the undesirability of matches. For example, all locations could contribute equally to a collective account that covers the estimated costs of integrating all refugees. Then, when a refugee is matched to a location, that location also receives the paired funds for integrating the refugee. Doing so will lead to a positive utility scenario and potentially to equal desirability of all non-returning agents.
>
> Altogether, our initial analysis seems to suggest efficient interventions will likely depend on the application domain and might require input from domain experts to assess their feasibility. Thus, in order to avoid giving a hasty suggestion, we concentrated on underlining the main causes for problems and kept the discussion on potential solutions at a high level. We left this interesting and important analysis for future work.

---

> ### Comment · Reviewer_EhdA · 2023-08-10
>
> I would like to.thank the authors for their response as I found.it interesting. I like the direction of this work  and continue to support this paper.

---

### Official Review · Reviewer_Zyns · 2023-07-06

**Soundness:** 4 excellent
**Presentation:** 4 excellent
**Contribution:** 3 good
**Rating:** 6
**Confidence:** 2

**Summary:**

This paper introduces a fresh perspective on attacks called *adversarial interaction attacks*, which can occur in markets that involve both a returning and a non-returning side. In such markets, agents' preferences are shaped by prediction mechanisms. While previous research has examined strategic behavior separately within the realms of matching and prediction mechanisms, the authors uncover the existence of these adversarial interaction attacks by explicitly considering the feedback loop between these mechanisms.

To investigate this phenomenon, the authors create a simplified setting and make several observations. They demonstrate that agents are driven by incentives to employ these attacks for personal gain. Moreover, the authors provide evidence that when returning agents adopt these attacks, it not only reduces the overall benefit but also amplifies inequality among non-returning agents.

**Strengths:**

**Significance:** This paper provides important insights into adversarial interaction attacks in markets with returning and non-returning sides. By shedding light on the underlying incentives and repercussions associated with these attacks, the authors deepen the understanding of strategic behavior within matching and prediction mechanisms. It also emphasizes the importance of developing strategies to mitigate these attacks and promote fairness in market outcomes.

**Novelty:** The attacks presented in this paper are novel. Previous studies have focused on strategic behaviors in either matching markets or prediction mechanisms individually, but this paper introduces a new perspective by examining the interplay between the two.

**Clarity and Quality:** The paper is well-written and comprehensible.

**Weaknesses:**

The setting analyzed is a bit too simplistic, perhaps this can be improved in the paper.

**Questions:**

N/A

---

> ### Author Rebuttal · Authors · 2023-08-08
>
> We thank Reviewer Zyns for their feedback on and engagement with our manuscript! We appreciate the positive feedback on the significance and novelty of our work and the expressed concern regarding the simplicity of our model.
>
> Regarding the latter, we tried to follow the model-simplicity principle introduced by Axelrod (1997) and widely used since. More precisely, our goal was to keep the model minimal in order to highlight the source of vulnerability. This also allowed us to create a general framework and, thus, show the feedback loop could be problematic in multiple settings.
>
> That being said, we agree that besides the advantages it provides, the simplicity of the model is also a limitation. As such, we reserved subsection 5.4 to discuss this choice and underline that a central direction for future work would be to consider more realistic models. As a first step, we used the appendix to extend the model (e.g., by considering more agents, alternative matching mechanisms, and k-NN as a more realistic recommender system) and perform simulations. This additional analysis largely confirmed our theoretical results. Most important, when there are few non-returning agents compared to available places and the utility of returning agents is always negative, adversarial interaction attacks continue to be effective for all analyzed market characteristics. While this analysis does not exclude the need for more realistic settings, we hope it provides some evidence that such attacks are worth investigating within various application domains.

---

> > ### Comment · Reviewer_Zyns · 2023-08-21
> >
> > Thanks for the response. I've read the response, and I've decided to keep my scores.

---

### Author Rebuttal · Authors · 2023-08-08

We would like to thank the organizers and reviewers for the opportunity to further discuss our work in this response phase and author-reviewer discussion! We are grateful for the extra work involved and will try to address the questions through direct responses.

---

### Decision · Program_Chairs · 2023-09-21

**Decision:**

Accept (poster)

**Comment:**

This paper looks at a dynamic two-sided matching model where (i) one side is single-shot and the other side returns each round and (ii) agents do not know the preferences up front but instead estimate/learn them via a predictor.  They show that strategic agents on the returning side can manipulate that predictor in the short term to potentially gain in the long term.  Strategic behavior in matching markets is not a new area of study, but the attack found in this model is (to the best of my/reviewers’ knowledge) new and interesting.  We encourage the authors to take the reviewers’ comments, and the authors’ own rebuttal, seriously to address remaining issues of motivation/generality.